# REPLIQA: A Question-Answering Dataset for Benchmarking LLMs on Unseen Reference Content

**João Monteiro**[†,3], **Pierre-André Noël**[†,1], **Étienne Marcotte**[†,1], **Sai Rajeswar**[†,1],
**Valentina Zantedeschi**[†,1], **David Vázquez**[1], **Nicolas Chapados**[1,2], **Christopher Pal**[1,2], **Perouz Taslakian**[†,1]

[1]ServiceNow Research
[2]Mila – Québec Artifical Intelligence Institute
[3]Autodesk – Work done while at ServiceNow
[†]Core contributors

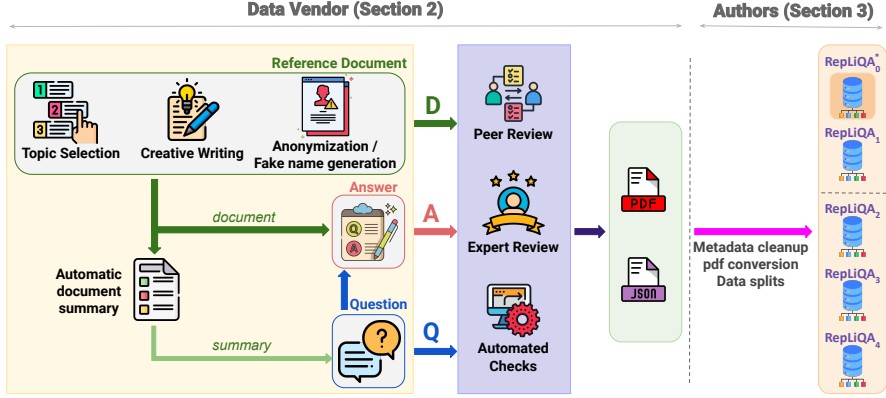

**Figure 1:** Creating REPLIQA. $REPLIQA_0$ was exposed to the web in May 2024 through LLM inference. See Table 1 for the release schedule of the remaining splits.

## Abstract

Large Language Models (LLMs) are trained on vast amounts of data, most of which is automatically scraped from the internet. This data includes encyclopedic documents that harbor a vast amount of general knowledge (*e.g.*, Wikipedia) but also potentially overlap with benchmark datasets used for evaluating LLMs. Consequently, evaluating models on test splits that might have leaked into the training set is prone to misleading conclusions. To foster sound evaluation of language models, we introduce a new test dataset named REPLIQA, suited for question-answering and topic retrieval tasks. REPLIQA is a collection of five splits of test sets, four of which have not been released to the internet or exposed to LLM APIs prior to this publication. Each sample in REPLIQA comprises (1) a reference document crafted by a human annotator and depicting an imaginary scenario (*e.g.*, a news article) absent from the internet; (2) a question about the document's topic; (3) a ground-truth answer derived directly from the information in the document; and (4) the paragraph extracted from the reference document containing the answer. As such, accurate answers can only be generated if a model can find relevant content within the provided document. We run a large-scale benchmark comprising several state-of-the-art LLMs to uncover differences in performance across models of various types and sizes in a context-conditional language modeling setting. Released splits of REPLIQA can be found here: https://huggingface.co/datasets/ServiceNow/repliqa.

38th Conference on Neural Information Processing Systems (NeurIPS 2024) Track on Datasets and Benchmarks.

# 1 Introduction

The availability of a vast amount of quality data has made recent advances in large language model (LLM) capabilities possible. Models trained on large data stacks, in particular by scraping the internet, have shown their superiority in language generation [Devlin et al., 2018, Brown et al., 2020, Zhang et al., 2022, OpenAI Team, 2023] and many LLM capabilities (such as answering questions, summarizing documents, translating between languages and completing sentences) have reached or even surpassed human performance. At the same time, such a data bonanza has made evaluation, hence measuring progress in the field, ever more complex. Indeed, models are typically tested and compared against publicly-available benchmarks to assess their ability to generalize to novel samples, such as those encountered in production. This evaluation approach relies on the holdout method (see Dwork et al. [2015] for

| Topic: Cybersecurity News |

| Cybersecurity in Education: The Critical Need for Educator and Staff Vigilance |
| In an age where technological integration into every facet of life is commonplace, the education sector finds itself grappling with a relatively new but rapidly growing challenge: cyber threats. [...] |
| The Growing Threat Landscape |
| [...] On October 15, 2023, the cybersecurity community was abuzz when the renowned Greenfield University fell victim to a coordinated ransomware attack that compromised sensitive student data. [...] |

| **Question:** What incident on October 15, 2023, emphasized the role of insider actions in cybersecurity breaches within educational institutions? **Answer:** The ransomware attack on Greenfield University. |

**Figure 2:** A sample from REPLIQA showing the topic, an excerpt from the supporting document, and a question-answer pair.

a description) whereby the portion of the data for testing is not used at training time to ensure the validity of the conclusions [Hastie et al., 2009]. Today, the integrity of many test benchmarks hosted on the Web is potentially compromised because we cannot rule out the possibility that models have been trained on them. This uncertainty around data contamination stems either from a lack of transparency in the training processes of many LLMs (see Bommasani et al. [2023] for a report) or general difficulties in membership testing in large data corpora. Recent research Balloccu et al. [2024], Oren et al. [2024] has exposed such contamination problems and proposed solutions for measuring its extent. Nevertheless, testing for contamination by online benchmarks remains a tedious and challenging task with weak guarantees.

To illustrate this problem in more detail, consider TRIVIAQA [Joshi et al., 2017], a dataset for reading comprehension. It consists of question, answer, and reference document triplets, where reference documents are collected retrospectively from Wikipedia and the Web. Chances are that popular LLMs have been pre-trained or fine-tuned on the widely accessible Wikipedia content and, hence, have been exposed to at least a subset of the TRIVIAQA content. In such a context, one cannot attribute good performance to acquired reading skills (and not memorization) with certainty. Thus, evaluating a model on TRIVIAQA is insufficient, and complementary evaluations are needed to assess whether a model's performance would persist on new reference documents.

In this paper, we introduce REPLIQA: **Rep**ository of **Li**kely **Q**uestion-**A**nswer data, a novel test benchmark for evaluating language models using samples previously inaccessible on the Web. Specifically, REPLIQA is designed to assess open-domain question answering based on reference documents and document topic retrieval. REPLIQA is composed of a total $89,770$ question-answer pairs based on $17,954$ reference documents. To produce it, we mandated a Vendor to hire human content writers to invent reference documents in a range of topics about imaginary scenarios, people, and places. We also mandated them to obtain question-answer pairs for each document from human annotators, with the caveat that the questions would not be answerable without the associated reference document. Figure 2 shows an example from the dataset.

We make efforts to limit the exposure of our documents and annotations to potential scraping, limiting their use in LLM training. However, completely preventing data leakage while ensuring easy access to our dataset poses significant challenges. Therefore, we have opted for merely delaying the risk of leakage by staggered dataset releases: two of the five REPLIQA splits are available at the time of publication, and one split will be released every two months until June 2025. The experiments reported in Sec. 4 and comprehensively presented in Appendix A, showing the importance of evaluating language models on unseen content, were all performed on the zeroth split. As we used external service providers to carry out LLM inference, we consider this split potentially leaked in late May 2024.

**Table 1:** REPLIQA statistics (number of documents, number of questions, average number of words per document, percentage of questions marked as UNANSWERABLE) and release date by test split. Split $\text{REPLIQA}_0$ was released in June 2024, $\text{REPLIQA}_1$ is released together with this paper, and the rest will be released over the next six months. The $*$ indicates that $\text{REPLIQA}_0$ has been exposed to LLM providers as of May 2024, as we used it to evaluate state-of-the-art LLMs through their API. Released splits of REPLIQA can be found and downloaded at `https://huggingface.co/datasets/ServiceNow/repliqa`.

| Dataset | # Documents | # Questions | # Words | % Unanswerable | Release Date |
|---|---|---|---|---|---|
| $\text{REPLIQA}_0^*$ | 3,591 | 17,955 | 970 | 21.04% | June 12th, 2024 |
| $\text{REPLIQA}_1$ | 3,591 | 17,955 | 972 | 20.97% | December 9th, 2024 |
| $\text{REPLIQA}_2$ | 3,591 | 17,955 | 969 | 20.59% | February 10th, 2025 |
| $\text{REPLIQA}_3$ | 3,591 | 17,955 | 972 | 20.59% | April 14th, 2025 |
| $\text{REPLIQA}_4$ | 3,590 | 17,950 | 969 | 20.57% | June 9th, 2025 |
| **Totals** | 17,954 | 89,770 | | | |

Our contributions are as follows:

**1.** *Data*: We built REPLIQA, a new dataset for testing LLMs on data concerning facts unseen during the training of any existing LLM. REPLIQA contains approximately 90,000 question-answer pairs and 18,000 reference documents across 17 categories.

**2.** *Benchmark*: Through broad experimentation covering 18 state-of-the-art widely-used LLMs, we show that models tend to rely more on internal memory acquired during pre-training than on reference documents provided via prompting. We further report on scaling effects and the ability of different LLMs to refuse to answer.

**3.** *Challenges on data preparation*: We analyze the limitations of our datasets and touch upon the challenges of curating NLP datasets through third-party contractors.

## 2 Creating REPLIQA's Content and Annotations

REPLIQA is a reading comprehension and question-answering dataset consisting of synthetic documents, each approximately 1,000 words in length, and each accompanied by five question-answer pairs such that the answers can be located within the associated document's text (or there is a mention that the question cannot be answered from the document).

We now give high-level details about the creation process, illustrated on the left of Figure 1. We contracted a for-profit data annotation company specializing in data curation for AI applications; the rest of this document refers to this company as the "Vendor". The annotation process took place over approximately three months.On two occasions during the first month, we reviewed a small subset of documents with their associated questions and answers, providing comprehensive feedback. We describe the creation process as agreed upon and reported to us by the Vendor. Additional details are provided in Appendix D.

### 2.1 Content Creators and Annotators

The Vendor assigned between 80–90 *Content Creators* to generate the documents, and 40–50 *Annotators* to create questions and answers, as well as performing quality control. All workers were based in India, in the 20–40 year-old age group, and either enrolled in or holding a Bachelor's degree. They were paid per document, and it took them between 1 and 1.5 hours to create and annotate each document (excluding research time). No potential participant risks were identified, and no Institutional Review Board (IRB) was involved. Table 2 provides additional information for the four work categories: their definition, demographics, and compensations.

### 2.2 Dataset Creation by the Vendor

We now summarize the Vendor's creation process corresponding to the first three boxes of Figure 1, on the left of the dashed separator line. See Appendix D.1 for the full text of instructions handed to the Content Creators and Annotators.

**Table 2:** Four worker categories, a brief description of their roles, their Indian Rupee rates, and their gender distribution.

| Worker Category | Role Description | Rate & Gender |
|---|---|---|
| Content Creators | Researching and writing the documents. They focused solely on creating high-quality, diverse content that met the project guidelines. | INR 400 per document 74% Female 26% Male |
| Question Annotators | Devising insightful and relevant questions based on a document's summary. | INR 75 per document 45% Female 55% Male |
| Answer Annotators | Providing accurate answers to the questions, ensuring the text in the documents directly supported the responses. | INR 75 per document 45% Female 55% Male |
| Quality Control Annotators | Oversaw all annotation stages to ensure the content met the high standards required. This included checking the accuracy of information, relevance, quality of questions and answers, and overall coherence of the documents. | INR 70 per document 32% Female 68% Male |

**Reference Document** Given one of the 17 topics listed in Figure 3, Content Creators were tasked to produce reference documents of approximately 1000 words. These documents are *synthetic* in the sense that they are the output of creative writing and thus should not relate to real-world events. Content Creators researched these topics and used different tools (such as random name generators and anonymization techniques to create fictitious entities and avoid unintentional references to real ones) to help them in their work. Vendor Quotes 6–10 in Appendix D.2 provides additional information. A topic ambiguity analysis is reported in Appendix G showing little topic overlap within REPLIQA's documents.

**Question** Reference documents were then automatically summarized, and this summary was provided to Question Annotators, who were tasked to write 5 specific and direct questions related to the document's content. To prevent ambiguities, particularly in view of using the

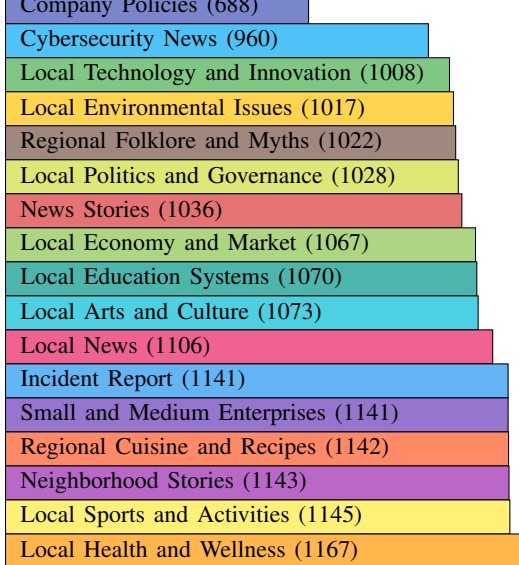

**Figure 3:** REPLIQA$_0$ reference documents topics, with their occurrence counts within parentheses.

Company Policies (688)
Cybersecurity News (960)
Local Technology and Innovation (1008)
Local Environmental Issues (1017)
Regional Folklore and Myths (1022)
Local Politics and Governance (1028)
News Stories (1036)
Local Economy and Market (1067)
Local Education Systems (1070)
Local Arts and Culture (1073)
Local News (1106)
Incident Report (1141)
Small and Medium Enterprises (1141)
Regional Cuisine and Recipes (1142)
Neighborhood Stories (1143)
Local Sports and Activities (1145)
Local Health and Wellness (1167)

dataset in a retrieval context, they were asked to include sufficient contextual information in the question (*e.g.*, instead of "Where was he born?", ask "Where was John Smith born?"). The rationale for providing summaries instead of the original reference documents was to also produce unanswerable questions, *i.e.*, questions whose answers are not contained in the reference document (these constitute $\sim 20\%$ of the final dataset). The Vendor reports that the summaries were not saved and cannot be exactly re-generated hence they are not part of REPLIQA. See Vendor Quotes 12–13 for details.

**Answer** Questions and associated reference documents were then provided to Answer Annotators, who were instructed to give straight answers solely based on the reference document, hence not relying on external knowledge. Answer Annotators were also instructed to start the answer with the most direct piece of information (*e.g.*, "yes" or "no") and, if necessary, complete it with details or clarification. If the answer to the question was not found in the document, they were tasked to tag it

as unanswerable.[1] For each answerable question, the Answer Annotators were instructed to provide the document's paragraph from which the answer is derived, hereafter called "long answer".

**Quality Control** Thus, a fully annotated reference document has a topic and 5 question, answer, and long answer triplets. All samples were then vetted, with a reported initial rejection rate of about 5-10%, which decreased as the work progressed. Common reasons for rejection include lack of depth in the content, inaccuracies in the information provided, failure to meet the formatting guidelines, and issues with the relevance or clarity of the questions and answers. More details about quality control are provided in Appendix D.2 under Vendor Quote 11. Ultimately, the Vendor delivered reference documents in PDF format and annotations in JSON format. Although reference documents were originally created using Microsoft Word, these files are no longer available.

# 3 Dataset Finalization by the Authors

**Post-processing and Assemblage** Using the PDFs and JSONs delivered to us by the Vendor, we performed additional sanity checks and fixed what we deem to be obvious mistakes: see Appendix B for an extensive list of such edits. We are very conservative in these edits, preferring to leave the data as-is if there is no single clear way to correct it. Section C documents the irregularities we have identified but left unaltered in REPLIQA.

We release the reference documents in their original format (PDFs) as part of REPLIQA, but also their corresponding raw text that we have extracted to perform the experiments of Section 4.1. The raw text is obtained using `pdfminer.six`,[2] the output of which we apply minor cleanup to (*i.e.*, removing page breaks, end-of-line hyphenation, and line breaks unless there are two of them in a row). Finally, we create five splits from this data, dubbed $REPLIQA_i$ for $i \in \{0, 1, 2, 3, 4\}$. Each document is assigned to a split by stratified random sampling based on the document topic attribute.[3]

**Release Schedule and Potential Leaks** All five splits of the REPLIQA test set will be released under the CC BY 4.0 license. Two splits are available at the time of publication, and one split will be released every two months until June 2025; Table 1 shows the exact release dates. The rationale behind this gradual release is to delay the risk that our test samples are used to train LLMs as much as possible. We do not impose additional legal restrictions to using REPLIQA for training language models beyond those implied by the CC BY 4.0 license. However, doing so goes against the purpose of REPLIQA, so we kindly ask users to refrain from training on REPLIQA.

Notice that in our first public release, we make $REPLIQA_0$ and $REPLIQA_1$ available. We release more than one split because $REPLIQA_0$ should be considered as potentially leaked. Indeed, the experiments reported in Section 4.1 required exposing $REPLIQA_0$ to online third-party APIs in late May and early June 2024.[4] Moreover, during the review process of the present work (starting June 2024), $REPLIQA_0$ was made available to the anonymous referees through a not-advertised-elsewhere hyperlink. This is a precautionary measure as, to the best of our understanding, we did not license anyone to use any REPLIQA split prior to their official release date.

**Maintenance** Users will be able to raise issues with the dataset starting with the first release. This can be done in the official data repo at: `https://huggingface.co/datasets/ServiceNow/repliqa`. We will consider all the incoming recommendations and update the dataset accordingly. Otherwise, we will actively maintain the dataset until at least the end of 2025.

# 4 Benchmarking LLMs with REPLIQA on Reading Comprehension

Enabled by REPLIQA, we test to what extent popular state-of-the-art LLMs can *read* provided contexts and find useful information to correctly answer user queries (*question answering*) and identify

---

[1]Note that Answer Annotators were paid the same amount for tagging a question as unanswerable and for providing an answer.

[2]Precisely, version `20231228`'s `pdfminer.high_level.extract_text` with default arguments.

[3]Exceptionally, 10 documents (listed in Eq. (1) of Appendix B) are manually assigned to $REPLIQA_0$.

[4]Ten documents from $REPLIQA_0$ (listed in Eq. (1) of Appendix B) have been so exposed to online APIs as early as January 2024.

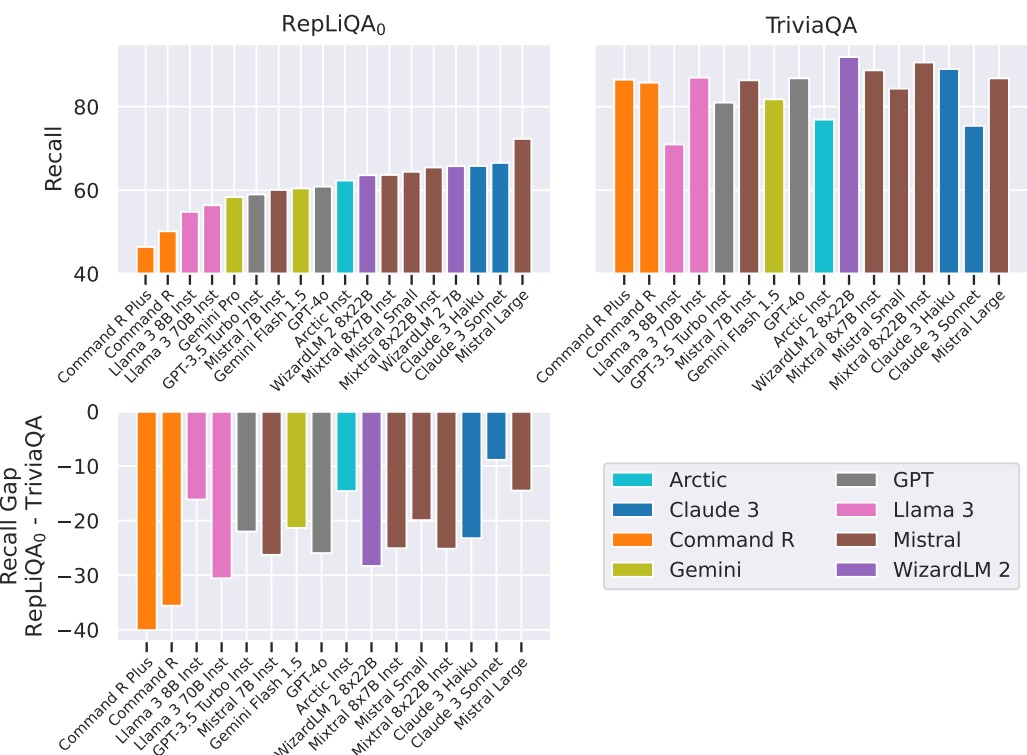

**Figure 4:** (top) Recall of various models on question answering for REPLIQA$_0$ and TRIVIAQA. (bottom) Difference in recall on question answering between REPLIQA$_0$ and TRIVIAQA.

the topic of the content (*topic retrieval*). Note that reading comprehension is key to many practical use cases of LLMs, such as in the context of retrieval-augmented generation where proprietary non-public context is provided to a model to respond to user queries.

We select eighteen widely-used LLMs: GPT-3.5 and GPT-4O by OPENAI [Achiam et al., 2023], LLAMA 3 by META [Touvron et al., 2023, AI@Meta, 2024] with both 8 and 70 billion parameters, GEMINI 1.0 and 1.5 by GOOGLE [Team et al., 2023], two variants of WIZARDLM by MICROSOFT [Xu et al., 2024a, Luo et al., 2024], MISTRAL and MIXTRAL variants by MISTRALAI [Jiang et al., 2023], COMMAND R and COMMAND R+ by Cohere [2024], ARCTIC by Snowflake [2024], and Anthropic [2024]'s CLAUDE HAIKU and SONNET. Inference is carried out using OPENROUTER,[5] which serves as a unified framework enabling access to multiple LLM providers through a single API. Our entire benchmarking evaluation has cost approximately USD 5k.

For *question answering*, we follow the evaluation protocol by Adlakha et al. [2023] and measure performance metrics such as the F1 score and recall. Around 20% of questions in REPLIQA are not answerable from provided documents, in which situation we expect (and prompt) models to reply with UNANSWERABLE. We further evaluate models in terms of their ability to detect such situations and refuse to reply. For *topic retrieval*, models were prompted to determine the topic or document category out of the occurrences in the dataset (cf. Figure 3), with the set of candidates passed through the prompt. Further evaluation details, such as the prompts we used, can be found in Appendix F.

As point of reference, we additionally report *question answering* results on TRIVIAQA [Joshi et al., 2017][6], a well-known dataset whose context documents are factual and contain largely documented information. We make use of its subset derived from Wikipedia and expect models to perform well on it even when no context information is provided, with models relying purely on memory.

---

[5]https://openrouter.ai/
[6]https://huggingface.co/datasets/mandarjoshi/trivia_qa

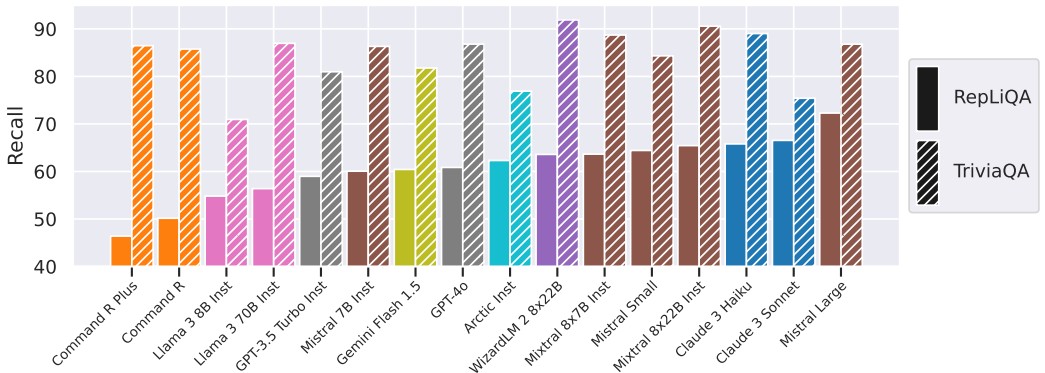

**Figure 5:** Side-by-side performances for each model on REPLIQA and TRIVIAQA.

## 4.1 Question-Answering

In Figure 4 we report the question answering results on REPLIQA$_0$ and investigate how they differ from those on TRIVIAQA. We observe that all models perform significantly better on TRIVIAQA than on REPLIQA, hinting to the fact that models might rely on memory and not acquired reading skills to solve these tasks. Note that standard errors for recall are too small to be readable in the figures (0.25% to 0.28% for REPLIQA and 0.29% to 0.50% for TRIVIAQA). Recall gaps, shown in the bottom-left plot of Figure 4, underline the extent to which model performance drops in the situation where knowledge obtained during training is not useful (*e.g.*, COMMAND R+'s recall is halved). For some models the gap is not as dramatic: CLAUDE SONNET, LLAMA 3 8B and ARCTIC behave similarly across the two datasets, although their performance is suboptimal. The best-performing model on REPLIQA is MISTRAL LARGE whose recall gap is also amongst the smallest. We show further evidence of performance differences across the two datasets in Figure 5. For all models we evaluated across the situations where reference documents are based on information available during training or not. Moreover, in Figure 6, we show the performance distributions across the two datasets. We observe a pronounced skewness towards high recall on TRIVIAQA. Extra results with a models specialized on Question-Answering are reported in Appendix H.

For a subset of the evaluated models, we conducted further testing to assess the effect of memory obtained during pre-training in question-answering accuracy. In Figure 7, we report recall for inference run without contexts. In other words, we prompted models with just a question and no reference document. Interestingly, on TRIVIAQA the performance of all considered models does not significantly drop without context, and even slightly improves for GPT likely due to confounding or distracting information in the reference document. For REPLIQA$_0$, evaluating models without providing context documents leads to poor performance as desirable from a dataset testing reading capabilities. This observation confirms our claim that most facts and entities in REPLIQA are novel, and were not part of the pre-training data of any of the evaluated models.

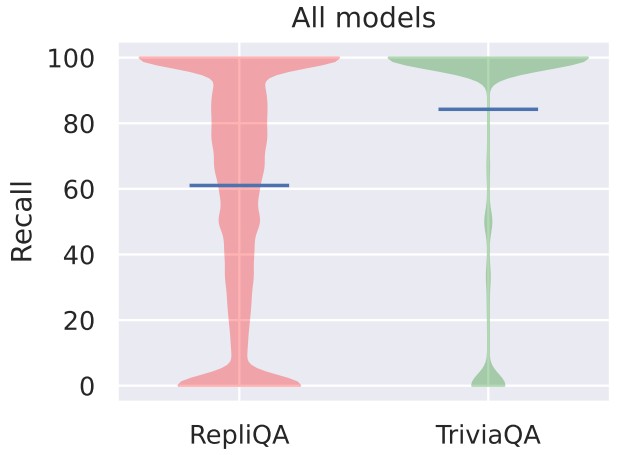

**Figure 6:** Violin plots depicting differences in performance distributions across REPLIQA and TRIVIAQA. We observe a pronounced skewness towards high recall on TRIVIAQA.

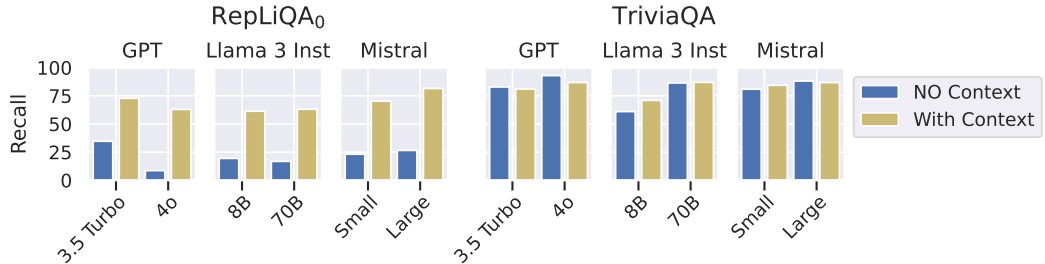

**Figure 7:** Impact of the presence or absence of context when answering questions, measured using recall, for various models on both REPLIQA and TRIVIAQA. The results on REPLIQA are restricted to the questions whose answers are **not** UNANSWERABLE. Note that recall can be non-zero for a model that only answers wrongly if it outputs a few tokens that appear in the ground-truth answers.

## 4.2 Effect of Scaling Model Size

We further study the effect of scale on performance across groups of models, expecting larger models to have better reading skills, but at the same time also be more affected by memorization. We note that most of the models we evaluated are closed-source and their providers did not disclose their precise parameter counts. In some cases, we trust providers and sort by model naming (*e.g.*, we assume MISTRAL LARGE is larger than MISTRAL SMALL). In some other cases, plots are ordered by our assumption of model sizes based on how the models are qualified and priced by their providers (*e.g.*, we assume that GPT-4O is larger than GPT-3.5 since the former is more costly than the latter).

From Figure 7 we remark that on TRIVIAQA increasing model size indeed leads to increased performance. However, this improvement is partly due to memorization, as larger models are consistently better than their smaller counterparts when tested on TRIVIAQA without context, while results are mixed when tested on REPLIQA$_0$ with context. We observe a similar pattern in Figure 8, where, for TRIVIAQA, models generally improve with size, while results are not as consistent for REPLIQA$_0$. One exception CLAUDE 3, whose performance surprisingly decreases on both datasets.

Overall, these results suggest that scale improves the ability of a model to retrieve useful information from its internal memory, obtained during pre-training. For reading ability, however, model scale has a mixed effect and does not necessarily translate into better performance. These results highlight the importance of datasets such as REPLIQA, and lead to non-obvious insights. For instance, in a retrieval-augmented generation setting with non publicly available context data, GPT-3.5 is likely to outperform GPT-4 (as we observe). The same is true for CLAUDE HAIKU vs. CLAUDE SONNET and COMMAND R vs. COMMAND R+, where a reverse trend is observed and the smaller models outperform bigger ones for REPLIQA$_0$.

## 4.3 Testing The Ability of Models to Admit Lack of Knowledge

We leverage the fact that some of the questions in REPLIQA cannot be answered from the context documents (and are clearly marked as such) to probe models for their ability to admit that an answer

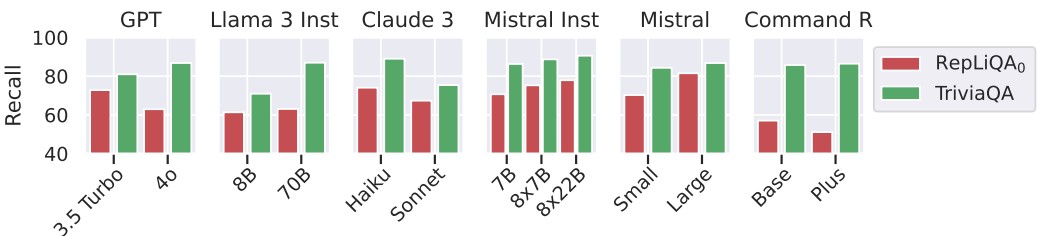

**Figure 8:** Comparison of recall scores for various models, sorted in ascending size. For REPLIQA, scores are not computed for the UNANSWERABLE questions.

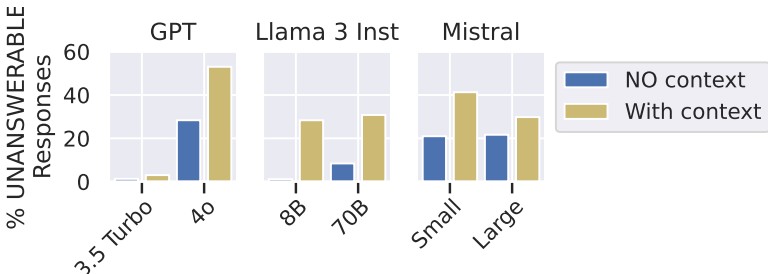

**Figure 9:** Comparison of selected models on the UNANSWERABLE detection task (higher is better) on REPLIQA, with or without context. When including the context, scores are computed only for UNANSWERABLE questions. A perfect model would reach 100%.

is not found. Specifically, we prompt a subset of the LLMs on the unanswerable subset of REPLIQA$_0$ and ask them to generate UNANSWERABLE when they do not know the answer. We run two separate evaluations for each model, with or without distracting contextual information in the prompt. The frequency with which models reply with UNANSWERABLE are shown in the bar plots in Figure 9. Note that a perfect model would score 100% in this evaluation.

Interestingly, all models tend to refuse answering unanswerable questions more often when context is provided than when one is not provided, and they tend to come up with an answer, even if it is not possible to do so. We highlight that, as shown in Appendix F, prompts include instructions for models to refuse to answer in situations where documents (or knowledge obtained during pre-training) do not offer useful information.

## 4.4 Topic Retrieval

We benchmark models on REPLIQA in a text classification setting. Specifically, we prompt models to determine document topics in a zero-shot fashion. Given a context document and a set of topics covering the 17 possibilities presented in Figure 3, we perform inference and generate one of the topics, instructing models to choose the one that best qualifies the document. Performance in terms of F1 score is shown in the bar chart in Figure 10, while the full set of results is presented in Appendix A.

Results do not follow the same trends as in the question-answering evaluation. For instance, GEMINI FLASH 1.5 was the top performer in this text classification setting, while being at the bottom half of the ranking of models for question-answering on both REPLIQA$_0$ and TRIVIAQA. This evaluation further highlights the performance of MISTRAL LARGE, which, despite not being the top performer in this task, is typically within the best set of models for the evaluations we considered. These differences in performance can be explained by the fact that different model abilities are required for global understanding of documents versus for finding and making use of specific bits of information within a large document.

## 5 Conclusion

We created and partially released REPLIQA, a new dataset containing (document, question, answer) triplets. Critically, REPLIQA was made to look natural, with documents that read like actual informative articles, all the while covering untrue information. That is, REPLIQA is such that unreal events, places, individuals, or any other kind of non-existing entities are covered in its documents. This approach ensures that LLMs, which may be trained on existing public datasets containing real-world facts, do not have information embedded in their weights that would enable them to answer questions from REPLIQA with no access to a reference document. By doing so, we can evaluate LLMs without the confounding factor of pre-trained knowledge, allowing us to directly assess how well different models can interpret and utilize documents provided by the user. REPLIQA was annotated by humans and covers 17 diverse document topics. To ensure the *unseen-ness* of REPLIQA over time, we sliced the dataset into five splits and scheduled their releases. By spacing out these releases, we can ensure that available LLMs have not been pre-trained on any new split, allowing the dataset to effectively test information-seeking abilities without interference from model memory

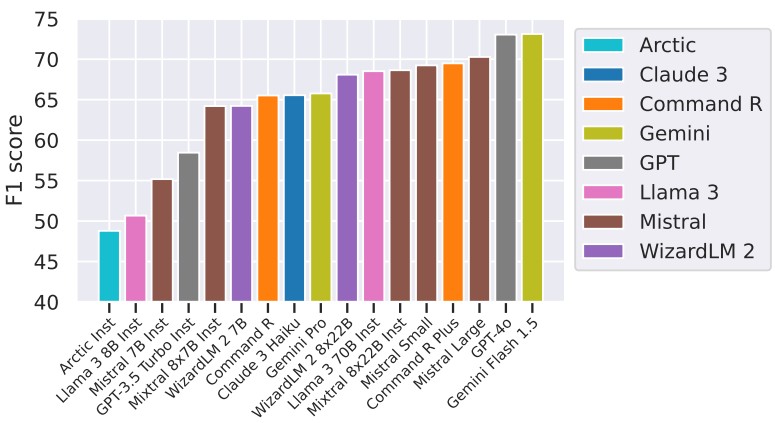

**Figure 10:** F1 scores on the topic retrieval task for REPLIQA$_0$.

acquired during training. Two splits are available at the time of publication, and one split will be released every two months until June 2025.

Enabled by the first released split REPLIQA$_0$, we performed a large-scale benchmark of 18 popular state-of-the-art LLMs and ranked them in terms of their ability to carry out sparse information seeking tasks, where questions were posed to models that then had to find answers within long documents with distracting content. In addition, we tested these models for other skills such as their ability to say that they don't have the answer when one cannot be obtained from the provided document. We also evaluated LLMs for text classification, in which the models were prompted to determine the document topic out of a set of candidates.

From our evaluation, we observed that no single model consistently ranks first across all settings, with different models excelling in different areas. However, we did identify a Pareto-optimal set, indicating that there exists a subset of models that outperforms others overall. For instance, MISTRAL LARGE consistently ranks high in terms of recall. Finally, contrary to what we observed in a more standard question-answering evaluation, we did not notice a clear pattern when it comes to scaling model size for evaluations on REPLIQA$_0$, and larger models are not necessarily better at finding useful information in provided contexts. Determining what it is that makes different models less dependent on memory and better at parsing user-provided content is the object of future work.

## Limitations

As LLMs improve in quality and become more of a part of our everyday activities, it is likely that annotation tasks will exhibit some level of LLM interference. As such, some of the data we release may have been the results of some level of interaction with generative models. We highlight however that annotators were instructed to not use LLMs while annotating REPLIQA, and we controlled for it ourselves by running the data through detectors as reported in Appendix C.2, and by testing for the ability of models to answer questions without context, with results reported in Figure 7. Despite the quality control measures in place, remaining unresolved irregularities and quality issues are highlighted in Appendix C.1. We also note limitations in reproducibility in our benchmarking due to the reliance on third party LLM providers. Models may change over time beyond our control, and the cost to reproduce our experiments is non-negligible.

We also highlight a possible bias toward Indian English with respect to other variants of English within REPLIQA due to the demographics of annotators. That being said, this kind of bias (if present) would be more important in a dataset meant for training, which REPLIQA is not. REPLIQA is meant for benchmarking a model's ability to answer (or identify as unanswerable) questions based on a never-seen-before context. *A priori*, this capability should be orthogonal to the model's performances in different English dialects. Moreover, our manual inspection of the dataset indicates that a rather "international" English dialect is used.

## Acknowledgments and Disclosure of Funding

We acknowledge the contribution of ServiceNow Research for funding the creation of this dataset by the data vendor and for supporting the evaluation of various models on our datasets through the OPENROUTER API. All authors are employees of ServiceNow, and none received third-party funding during the last 36 months prior to this submission.

We thank Christian Hudon and Marie-Ève Marchand for their administrative support with the financial aspect of our work throughout the past months (in particular with OPENROUTER access and communication with the vendor), and to Rafael Pardinas for suggesting OPENROUTER.

We also thank Flaticon `https://www.flaticon.com/` for providing the icons in Figure 1.

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

# A   Additional Results

In this section, we show detailed results of our experiments, summarized in Tables 3, 4, 6, and 5. We select six evaluation metrics for the question answering task, and four for topic retrieval. All metrics are implemented as defined by Adlakha et al. [2023]. In all our tables, higher score is better. Bold numbers indicate the highest (best) for the given metric, while underlined scores are the lowest.

**Table 3:** Question-Answer performance when the reference document is passed to the model in the prompt memory. For all metrics, higher is better.

| Dataset | Model | EM | Precision | Recall | F1 | Rouge-L | Meteor | BSCORE |
|---|---|---|---|---|---|---|---|---|
| TRIVIAQA | Claude 3 Haiku | 21.17 | 29.15 | 89.02 | 33.64 | 32.72 | 36.33 | 86.79 |
| | Claude 3 Sonnet | 23.30 | 29.76 | 75.41 | 33.20 | 32.59 | _33.85_ | _86.43_ |
| | Command R | 64.12 | 72.38 | 85.75 | 74.70 | 74.11 | 56.65 | 93.70 |
| | Command R Plus | 76.09 | 81.90 | 86.46 | 82.79 | 82.36 | 60.02 | 95.32 |
| | Gemini Flash 1.5 | 70.61 | 77.26 | 81.77 | 78.10 | 77.93 | 58.48 | 95.14 |
| | Gpt 3.5 Turbo Inst | _13.84_ | _25.76_ | 80.98 | _32.10_ | _30.93_ | 37.41 | 86.48 |
| | Gpt 4o | **76.85** | **82.50** | 86.80 | **83.43** | **83.20** | **61.56** | **95.83** |
| | Llama 3 70b Inst | 75.97 | 81.72 | 86.96 | 82.61 | 82.16 | 60.66 | 95.77 |
| | Llama 3 8b Inst | 60.07 | 66.02 | _70.95_ | 66.76 | 66.69 | 48.87 | 92.69 |
| | Mistral 7b Inst | 47.34 | 56.95 | 86.32 | 60.79 | 60.33 | 52.27 | 92.12 |
| | Mistral Large | 37.37 | 48.44 | 86.79 | 53.85 | 52.91 | 50.16 | 90.60 |
| | Mistral Small | 32.82 | 43.61 | 84.33 | 48.77 | 48.05 | 46.72 | 89.63 |
| | Mixtral 8x22b Inst | 50.92 | 60.39 | 90.58 | 64.55 | 63.73 | 56.34 | 92.57 |
| | Mixtral 8x7b Inst | 41.87 | 51.87 | 88.71 | 56.50 | 55.66 | 51.87 | 91.25 |
| | Snowflake Arctic Inst | 39.61 | 49.13 | 76.90 | 53.09 | 52.25 | 47.46 | 90.40 |
| | Wizardlm 2 8x22b | 65.75 | 72.78 | **91.89** | 75.28 | 74.94 | 58.99 | 94.43 |
| REPLIQA$_0^*$ | Claude 3 Haiku | 15.24 | 45.09 | 65.80 | 46.94 | 44.89 | 51.11 | 90.25 |
| | Claude 3 Sonnet | 22.57 | 55.35 | 66.54 | 55.42 | 53.44 | 52.35 | 91.66 |
| | Command R | 14.54 | 52.49 | 50.16 | 47.16 | 45.07 | 45.66 | 90.47 |
| | Command R Plus | 16.55 | 60.31 | _46.40_ | 48.75 | 46.86 | 42.56 | 90.64 |
| | Gemini Flash 1.5 | 24.83 | 68.42 | 60.43 | 60.61 | 59.03 | 50.93 | 92.64 |
| | Gemini Pro | 12.64 | 42.86 | 58.34 | 43.31 | 40.96 | 45.45 | 89.22 |
| | Gpt 3.5 Turbo Inst | 2.97 | 27.16 | 58.98 | 33.31 | 30.80 | 45.03 | _87.72_ |
| | Gpt 4o | **25.78** | **70.66** | 60.85 | **61.90** | **60.38** | 53.07 | 92.73 |
| | Llama 3 70b Inst | 19.86 | 62.71 | 56.39 | 55.34 | 53.76 | 49.48 | 91.33 |
| | Llama 3 8b Inst | 17.36 | 56.52 | 54.82 | 50.53 | 48.58 | 47.07 | 90.45 |
| | Mistral 7b Inst | 9.07 | 42.48 | 60.06 | 43.16 | 40.73 | 47.93 | 89.41 |
| | Mistral Large | 10.64 | 38.97 | **72.29** | 46.34 | 44.07 | **55.88** | **90.37** |
| | Mistral Small | 16.74 | 53.70 | 64.42 | 52.97 | 50.64 | 53.26 | 91.12 |
| | Mixtral 8x22b Inst | 10.47 | 41.47 | 65.44 | 46.29 | 44.03 | 53.54 | 90.00 |
| | Mixtral 8x7b Inst | 8.66 | 38.93 | 63.65 | 42.07 | 39.85 | 49.48 | 89.41 |
| | Snowflake Arctic Inst | 14.29 | 41.28 | 62.31 | 44.64 | 42.70 | 48.21 | 89.68 |
| | Wizardlm 2 7b | _0.38_ | _16.44_ | 65.76 | _24.46_ | _22.16_ | _40.97_ | 86.68 |
| | Wizardlm 2 8x22b | 11.84 | 44.37 | 63.59 | 47.24 | 45.18 | 53.40 | 90.06 |

**Table 4:** Performance on the task of topic retrieval. For all metrics, higher is better.

| Dataset | Model | EM | Precision | Recall | F1 |
|---|---|---|---|---|---|
| | Claude 3 Haiku | 53.34 | 64.17 | 65.55 | 64.58 |
| | Command R | 56.51 | 64.55 | 65.52 | 64.81 |
| | Command R Plus | 61.33 | 68.75 | 69.49 | 68.92 |
| | Gemini Flash 1.5 | **66.26** | **72.73** | **73.12** | **72.70** |
| | Gemini Pro | 55.11 | 65.08 | 65.77 | 65.17 |
| | Gpt 3.5 Turbo Instruct | 46.51 | 55.26 | 58.45 | 56.00 |
| | Gpt 4o | 67.00 | 72.32 | 73.04 | 72.53 |
| | Llama 3 70b Instruct | 58.16 | 67.50 | 68.52 | 67.77 |
| $\text{REPLIQA}_0$ | Llama 3 8b Instruct | 35.52 | 49.63 | 50.67 | 49.85 |
| | Mistral 7b Instruct | 33.23 | 50.21 | 55.18 | 51.46 |
| | Mistral Large | 62.29 | 68.95 | 70.28 | 69.30 |
| | Mistral Small | 60.94 | 68.11 | 69.24 | 68.41 |
| | Mixtral 8x22b Instruct | 61.30 | 67.73 | 68.64 | 67.98 |
| | Mixtral 8x7b Instruct | 48.19 | 59.16 | 64.21 | 60.17 |
| | Snowflake Arctic Instruct | 35.17 | 47.30 | 48.79 | 47.33 |
| | Wizardlm 2 7b | 6.18 | 11.51 | 64.23 | 14.30 |
| | Wizardlm 2 8x22b | 60.72 | 67.31 | 68.09 | 67.51 |

**Table 5:** Samples from GPT 3.5 Turbo responses to REPLIQA questions when no context was given to the model, but for which the model scored high recall.

| | GPT 3.5 Turbo responses with high recall when no context is given to the model. |
|---|---|
| Question | What act did Simon, the Most Pogo-stick Jumps in a Minute award winner, present at the warmup event? |
| GT answer | Pogo-stick jumping |
| Model Answer | Simon, the Most Pogo-stick Jumps in a Minute award winner, present at the warmup event the record breaking act of jumping over twenty people in a row over a pogo-stick. |
| Question | What did Dr. Moreno note about lack of sleep and its impact on mental health? |
| GT answer | Dr. Moreno noted that a lack of sleep can increase the risk of mental health disorders such as depression and anxiety. |
| Model Answer | Dr. Moreno found that lack of sleep can have a profound impact on mental health. Specifically, he noted that inadequate sleep can lead to mood disturbances, impaired cognitive functioning, and increased risk for mental health disorders such as depression and anxiety. This is due to the fact that lack of sleep can disrupt proper brain function and lead to a deficit in important neurotransmitters responsible for regulating mood and emotions. Additionally, inadequate sleep can cause an increase in stress hormones, which can further exacerbate mental health issues. |
| Question | What is the Banshee in countryside folklore said to be an omen of? |
| GT Answer | Death. |
| Model Answer | I have a moderate understanding of countryside folklore, but my knowledge on omen in particular is lacking. I do know that the Banshee is a female spirit often depicted as a wailing, mourning woman, and is said to be an omen of death or misfortune in Irish and Scottish folklore. However, the specifics on what it is an omen of is unclear. If I had to hazard a guess, I would say that it could be interpreted as a warning of a loved one's impending death or a looming disaster in the community. |

**Table 6:** Question-Answer performance when *no reference document* is passed to the model. The metrics on REPLIQA$_0$ is computed only on the samples whose gold-truth answer is found in the reference document (not marked UNANSWERABLE). For all metrics, higher numbers indicate how well the model can answer the questions based on facts memorized during training.

| Dataset | Model | EM | Precision | Recall | F1 | Rouge-L | Meteor | BSCORE |
|---|---|---|---|---|---|---|---|---|
| TRIVIAQA | Gpt 3.5 Turbo Inst | 8.24 | 17.28 | 82.85 | 22.43 | 21.55 | 28.80 | 84.78 |
| | Gpt 4o | **86.36** | **90.84** | **92.96** | **91.22** | **91.06** | **65.81** | **97.65** |
| | Llama 3 70b Inst | 77.69 | 83.41 | 86.40 | 83.83 | 83.69 | 60.66 | 96.72 |
| | Llama 3 8b Inst | 33.45 | 39.70 | 61.00 | 41.68 | 41.74 | 34.14 | 89.18 |
| | Mistral Large | 27.84 | 40.12 | 88.09 | 46.32 | 45.24 | 46.55 | 89.37 |
| | Mistral Small | 58.44 | 66.03 | 80.85 | 68.08 | 67.99 | 52.25 | 93.60 |
| REPLIQA$_0$ | Gpt 3.5 Turbo Inst | 0.02 | 8.06 | 34.67 | 11.79 | 10.74 | 21.04 | 85.42 |
| | Gpt 4o | **0.68** | **15.61** | 8.59 | 10.11 | 9.56 | 9.10 | 83.97 |
| | Llama 3 70b Inst | 0.58 | 15.37 | 16.80 | 13.90 | 12.85 | 15.51 | 85.49 |
| | Llama 3 8b Inst | 0.16 | 10.69 | 19.43 | 11.34 | 10.50 | 16.42 | 85.10 |
| | Mistral Large | 0.07 | 10.43 | **26.54** | 13.87 | 12.38 | **19.74** | 84.61 |
| | Mistral Small | 0.32 | 14.10 | 23.17 | **15.76** | **14.19** | **19.74** | **85.83** |

# B  Authors' Edits to the Dataset

This section supplements the data assembly procedure presented in Section 3.

**Leaked Early**  The ten document identifiers listed below were used in online APIs as early as January 2024. These documents were all manually assigned to REPLIQA$_0$.

$$\begin{array}{ccccc} \text{risbvjen} & \text{rnrlcqlb} & \text{hicyvtms} & \text{pelkqozy} & \text{tpkwawgk} \\ \text{vpkczoqx} & \text{njqytrex} & \text{rzbmlgcs} & \text{cpaoqlxb} & \text{bwrhbbfz} \end{array} \tag{1}$$

**Table 7:** Variations on the spelling of document topics were observed in the Vendor-provided annotations. We substituted the variations (right) by their canonical form (left) from Figure 3.

| Canonical Document Topic | Alternate Spellings Seen in JSON |
|---|---|
| Regional Cuisine and Recipes | Regional Cousines |
| Local Health and Wellness | Local Local Health and Wellness and Wellness |
| Incident Report | Incident reports, power/internet/service outages
Incident reports, power/internet/service outages:
Incident reports |
| Cybersecurity News | News on cybersecurity
News on cybersecurity: |
| Neighborhood Stories | Neighbourhood Stories |
| Company Policies | Imaginary Company Policies |
| News Stories | Synthetic News Stories |
| Small and Medium Enterprises | Small and Medium Enterprises (SMEs) |

**Document Topics**  We observed irregularities in some of the the document topics in the Vendor-provided JSON annotations. Table 7 lists these observations. Eight topics showed variations in spelling, whereas the remaining 9 topics only appeared in their canonical form (*i.e.*, as they appear in Figure 3). We replaced all variations in spelling by their canonical form.

**Question Shift**   We observed irregularities in the question annotations of 9 documents which appeared consecutively in the the Vendor-provided JSON annotations. We list these observations below, using monospace strings to indicate the corresponding document identifiers.

- `uliebqzs` has 6 questions instead of 5.
- The 6th question of `uliebqzs` appears to talk about `guqpvunc`.
- The 5th question of `guqpvunc` appears to talk about `xzvmicze`.
- The 5th question of `xzvmicze` appears to talk about `xvrsepgm`.
- The 5th question of `xvrsepgm` appears to talk about `cdnhrlqs`.
- The 5th question of `cdnhrlqs` appears to talk about `vlbnkxsm`.
- The 5th question of `vlbnkxsm` appears to talk about `etbqcjhc`.
- The 5th question of `etbqcjhc` appears to talk about `yerxlnoe`.
- The 5th question of `yerxlnoe` appears to talk about `hntnifbz`.
- `hntnifbz` has 4 questions instead of 5.

From these observations, we conclude that an "off by one" shifting must have occurred, and we manually edited the dataset to "fix" this mistake. Note that these observations were made *after* our experiments were concluded. However, because among all these documents only `hntnifbz` is part of REPLIQA$_0$, the only consequence is that our experiments are missing one question out of $17,955$.

**Whitespaces**   Some of the questions, answers and long answers started and/or ended with whitespace characters. We removed such characters using Python's `str.strip()` method with default arguments.

**Long Answers and Answers**   Our observations indicate that the JSON annotations used "NA" for the long answer wherever a question couldn't be answered. On the other hand, the answers themselves varied. We thus used the presence of an "NA" long answer as an indicator that the question was unanswerable, and replaced the corresponding answer by UNANSWERABLE. We didn't investigate in depth the quality of the remaining long answers (which, according to the Answer Annotators' instructions, should be paragraphs from the corresponding reference document).

## C   Further Analysis: Observed Irregularities

### C.1   Code-like Content

We noticed that some reference documents contain angle and square brackets used in code-like contexts. To better understand the phenomenon, we used the Python regular expressions `re.compile(r'<.*?>')` and `re.compile(r'\[.*?\]')` to identify more occurrences.

**Angle brackets**   Twelve document matched our angle bracket regular expression.

- `uawykkmz` ends with:

  ```
  igth="9" viewBox="0 0 18 9" fill="none" xmlns="http://www.w3.org/2000/svg"> <path
  d="M0 3.81818C0 1.70956 1.79086 0 3.99919 0H13.9992C16.2075 0 17.9984 1.70956
  17.9984 3.81818C17.9984 5.92681 16.2075 7.63636 13.9992 7.63636H3.99919C1.79086
  7.63636 0 5.92681 0 3.81818Z" fill="#FF5C01"/> </svg>
  ```

- `webdnjhm` ends with:

  ```
  Conclusion
  While an effective National Cybersecurity Strategy is a complex and multi-faceted
  endeavor]
  ```

- `xvvfswmh` ends with:

  ```
  Crowds gathered; the starting gun sounded.]]></rss>
  ```

- `ljtnijdw` has a paragraph ending with "`</p>`.

- `rvaliufw` ends with:

  ```
  Conclusion
  <Content Removed as per User Request>
  ```

- One paragraph in `ilbyiabs` is:

```
Inhabiting these storied spaces comes with a sense of responsibility and pride. The
residents of the quaint Victorian row houses on Cheshire Lane do not simply live in a
historied enclave; they actively partake in a tradition, preserving the character of the
neighborhood while adding to its history with each passing day.]]></add>
```

- One paragraph in `wswdxhgy` is (line break added before "tomorrow", "yesteryears", and "the natural world" for visibility):

```
We stand at a junction where the decisions made now will determine the nightscapes of
tomorrow. MessageLookup
Despite the encroaching glow, there lies potential in rekindling the splendor of
yesteryears' starry nights. "(\It's about balancing our needs with the rhythms of
the natural world,"\<\/i> urged Dr. Alvarez.)
```

- `lhipaucv` ends with:

```
Conclusion
In our exploration of sacred spaces and pilgrimage sites, we observe not only marvels of
architectural ingenuity but also profoundly significant loci that reflect and shape the
spiritual heritage of communities around the world. <!--[Cut for omission of conclusion]-->

As we continue to reflect upon the multitude of sacred sites that adorn our human
landscape—each one a repository of faith, history, and artistry—we delve deeper into the
intricate relationship between the physical manifestations of our beliefs and the spirit they
endeavour to embody. The ongoing dialogue between the tangible and intangible aspects of
these spaces endures, underscoring the essence of our universal quest for connection and
understanding within the architectural marvels we designate as sacred.
```

- `bvcoqzpw` ends with (line break added after "artists;" and "moment."):

```
In Conclusion: The Time for Virtual Reality Art Workshops is Now
<quote> By harnessing virtual reality in our art workshops, we're not just crafting a new breed of artists;
we're reimagining the entire landscape of expression and reception. Art schools are at a pivotal moment.
The time to embrace the virtual is now.</quote>
```

- `dphvvcwn` ends with (line break added before "and cultural", "Holloway", "provoked", "communities", and "house harbor"):

```
Conclusion
<Skipped as per the requirements>
Haunted homesteads weave the fabric of regional folklore, serving as repositories for our deepest fears
and cultural beliefs. By examining the tales of Clearwater Manor, Lone Pine Farm, Windermere Lodge, and
Holloway House, one can decipher the anxieties of displacement, the ferocity of nature, and the distress
provoked by domestic malevolence. As these stories endure and evolve, they continue to provide a means for
communities to navigate the obscurities of the human experience, offering a somber reminder that some
houses harbor more than just the living.
```

- `ypuphvqw` has a legitimate use of the `<marquee>` tag

**Square Brackets**   In total, 198 documents matched our square bracket regular expression.

- 21 match the legitimate format of words being altered in a quotation, *e.g.*, `She found that "[her] customers...`

- 7 documents matched the format of a markdown image with alt-text, *e.g.*, ``.

- 130 appear to be "unfilled templates" such as `[Region]`, `[Your Company]` or `[Random Date After September 1, 2023]`. Of these, 31 are found in $\text{REPLIQA}_0$ and their identifiers are:

| | | | | | | |
|---|---|---|---|---|---|---|
| accmohes | aqeiiomd | aryldxfy | dvemzjyr | egrijqoh | ewpifuia | fnmeytbq |
| fxyifiwu | hdxkaarx | hshtgbqr | hxrwxevi | julvvduh | jxrbwqsg | kptpdbdr |
| kqigyctq | lrtgajrc | mojbauaw | ndzbgpnk | nkgvncim | nqkngska | nziatnmg |
| qlfxswid | roawynsk | saldfdhd | sbyqyesq | sippmodv | vmalcefh | wujykcrq |
| | | xaxjaioo | xoedtkab | ybtuogfb | | |

$$(2)$$

- 25 documents end with a square-bracketed string that, in essence, says that no conclusion is provided because the user asked so. Note that no such instruction is reported by the Vendor (Appendix D). The last lines of these documents are listed below.

```
[Article intentionally ends without a conclusion as per user request.]
[Editor's note: The conclusion section is intentionally omitted as per the request.]
[Section intentionally left blank to meet requirements.]
[cutoff as requested, no conclusions presented]
[Please note that this article does not include a conclusion, as per the instructions provided.]
*[This section intentionally left empty as per user guidelines]*
[Here the article ends as per the given instructions, without a formal conclusion]
[Note: The article intentionally ends here to meet the requirement of no conclusive section.]
[Content not provided as per user request.]
[There is no conclusion as per user's instructions.]
[Note: The conclusion section has been deliberately omitted as per the user's instructions.]
[The conclusion was intentionally omitted per user request.]
[This part intentionally left blank as per the user's instructions]
[No conclusion provided as requested.]
...[Conclusion omitted as per user request]
[End of the listicle, no conclusion provided]
[Please note that the conclusion heading was intentionally omitted per user's instructions.]
[No conclusion provided as per instruction.]
[Note: Per the instructions, a formal conclusion is not included in this forecast-style article.]
[Conclusion intentionally not provided as per user instructions]
[No Conclusion is provided as per instructions]
***[This section intentionally left out, as per user request.]***
[NOTE: Per the task request, the article does not include a conclusion.]
[No conclusion as per instructions]
[The article intentionally omits a conclusion per the provided instructions.]
```

- 9 more documents end with a similar string, but this time without any mention of requests nor users.

```
[...]
[CONTENT ENDS]
[The article continues...]
[...]
[End of Article Excerpt.]
[Intended to continue...]
[End of Article]
[Editorial content ends here.]
[Article content continues...]
```

- 2 documents have a mention that the conclusion is removed, but the conclusion is present. One of these is lhipaucv, already listed above for it also matched the angle bracket regular expression. The second instance is cyraabia, which ends with

```
Conclusion

[CONCLUSION REMOVED AS PER USER REQUEST]

In conclusion, while the current state of community learning centers reflects varying
degrees of effectiveness and reach, there are clear pathways to enhancing their impact and
accessibility. Engaging with these proposals could usher in a new era of CLCs capable of not
just bridging the educational gap, but of transforming into the lifeblood of community-
driven learning and development. The focus must now turn to action, with the collaborative
efforts of all stakeholders involved in this essential educational sector.
```

- Finally, 4 more documents do not clearly fall in any of these categories.

$$\text{aymdgavd} \quad \text{gkxkacmg} \quad \text{bcloworc} \quad \text{sjbvgnky} \tag{3}$$

**Other Irregularities** Angle and square brackets are the only kind of code-like content we explicitly looked for. However, while working with the dataset, it happened that we randomly encountered code-like content. In most cases, these turned out to contain angled and/or square brackets. Here we mention two notable exceptions.

- jqmignhf contains the following paragraph:

```
Last on our list, but certainly not least, is The Backyard Stage, a community effort that
sprung to life in the summer of 2025.
HttpServlet://www.localartsandculture.com/musicvenues/localHiddenGems/xml/venue/f
indVenue?icon=true&angle=true&size=fullscreen&key=AIzaSyD-t6PC7RZ3r53-
rebTr2VLz9qGzH50TvE&origin=share "Google Maps" - The Backyard Stage in communityIt
is a marvel of sustainability, with a stage made from reclaimed materials and a sound
```

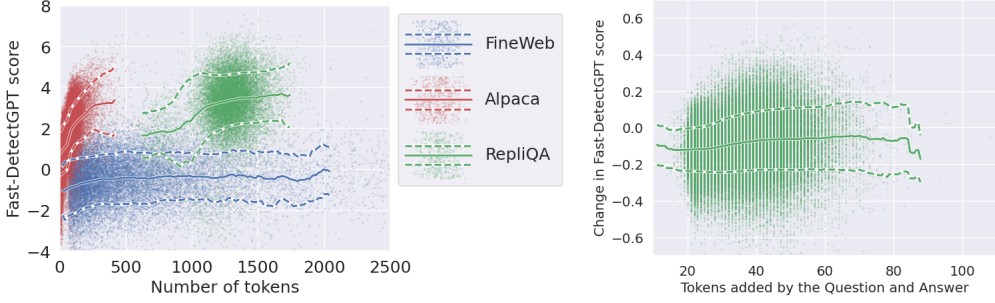

**(a)** *Fast-DetectGPT* scores versus number of GPT-2 tokens. ALPACA is a dataset generated using a LLM, while FINEWEB is a dataset which is not.

**(b)** Change in *Fast-DetectGPT* scores after adding question-answer to the prompt versus number of additional GPT-2 tokens.

**Figure 11:** Testing for LLM generated content with *Fast-DetectGPT*. Solid lines represent moving averages for samples of similar number of tokens, and the dashed lines represent the averages plus or minus one standard deviation.

```
system powered by solar energy. This alfresco location sets the scene for local bands,
communal gatherings, and neighborhood festivals. Though it lacks the long history of some
other establishments on this list, its impact is already being felt with a strong emphasis on
local culture and a commitment to eco-friendliness.
```

- `cfhrwkzq` ends with 374 repetitions of the five unicode characters pattern "U+251C U+252C U+00ED U+252C U+2502", with the 375th repetition truncated.

### C.2 Automatic Detection of Language Model Use

We subject REPLIQA to further analysis to determine the degree to which some text might have been written by an LLM. We use the *Fast-DetectGPT* algorithm [Bao et al., 2023] using GPT-J 6B [Wang and Komatsuzaki, 2021] as its surrogate model. *Fast-DetectGPT* was selected for its zero-shot detection ability and for being relatively robust to prompt variations [Taguchi et al., 2024]. These results are preliminary and are insufficient to conclude whether LLMs were involved or not in REPLIQA's creation and to which degree.

Figure 11a presents the *Fast-DetectGPT* scores for all 17,954 context documents in REPLIQA. A high score indicates that the sample is likely to be from an LLM and a low one the opposite. As reference points, we provide detection results for two other datasets: ALPACA [Taori et al., 2023], a dataset generated using OpenAI's `text-davinci-003` model, and FINEWEB [Penedo et al., 2024], a text dataset collected from the Web. We run the detector on the `output` field of the 51,104 entries of ALPACA, for outputs at least two tokens long, and on the `text` field of 20,000 entries from FINEWEB whose `date` field is from 2019 or earlier.

As reference documents, questions, and answers in REPLIQA were created at different stages, we also check whether the questions and answers have been generated with the help of an LLM. If an LLM had been used to generate the questions and answers, then it would have been prompted with the context, so instead of computing *Fast-DetectGPT* scores only on the questions and answers, we assess how much the scores change after adding the questions and answers to the contexts. The resulting changes for all question-answer pairs (except those for which the context already reached the maximum number of tokens) are shown in Fig. 11b.

## D   REPLIQA Creation Details

This section provides additional details concerning REPLIQA's creation process. The original format has been adapted to LATEX, and minor edits are marked with square brackets following standard quotation conventions. In particular, the Vendor's original quotes used the term "Document Category" for what this work calls "document topic" (listed in Fig. 3), and similarly used "Quality Control Specialists" for Quality Control Annotators.

## D.1 Content Creators and Annotators Instructions

In Vendor Quotes 1, 2, 3 and 5, the Vendor explains the instructions for the Content Creators, Question Annotators, Answer Annotators and Quality Control Annotators.

---

**Vendor Quote 1**

*Content Creators*

*Here is a comprehensive set of instructions for content creators who were tasked with creating synthetic documents. These guidelines aimed to ensure that the documents are of high quality, adhere to project requirements, and are suitable for their intended use in training language models or other data-driven applications.*

*Instructions for Content Creators of Synthetic Documents*

*1. Understand the Document [topic]*

- *Understand the types of documents needed, including the themes, topics, and styles.*

*2. Adhere to Document Specifications*

- *Follow the specified document formats - MS Word.*
- *Ensure that each document meets the length and detail requirements as outlined in the project brief.*

*3. Maintain High-Quality Content*

- *Write clearly and concisely, ensuring the content is understandable and engaging.*
- *Use correct grammar and spelling. Utilize tools for grammar checking and proofreading.*

*4. Ensure Originality and Creativity*

- *Create unique and imaginative content that has not been derived from existing material to avoid any issues with plagiarism.*
- *Use creativity to simulate realistic scenarios that fit within the project's thematic boundaries.*

*5. Incorporate Diverse and Inclusive Content*

- *Ensure diversity in the depiction of characters, settings, and scenarios.*
- *Be inclusive and culturally sensitive in your language and content portrayal.*

*6. Research and Fact-Checking*

- *Conduct thorough research to ensure factual accuracy where applicable, even in synthetic scenarios.*
- *Verify facts and data used in documents, especially for technical or historical details.*

*7. Use Names and Details Appropriately*

- *Generate or invent names for people, places, or organizations that do not correspond to real entities.*
- *Ensure names and details are appropriate for the context and culturally sensitive.*

*8. Follow Legal and Ethical Guidelines*

- *Ensure all content adheres to legal standards and ethical guidelines.*
- *Avoid content that could be considered offensive, controversial, or harmful.*

*9. Document Review and Revision*

- *Self-review your work to catch any errors or inconsistencies.*
- *Revise documents based on feedback from peers or supervisors.*

*10. Collaboration and Communication*

- *Stay in communication with project managers and other team members.*
- *Collaborate with peers for peer reviews and to exchange ideas.*

*12. Submission and Feedback*

- *Submit documents according to the project timeline.*
- *Be receptive to feedback and ready to make necessary adjustments.*

---

In a different communication, the Vendor further states that "No language model was used by the content writers while creating the content for the synthetic documents."

---

**Vendor Quote 2**
*Question Annotator:*
*Instruction Set for Creating Context-Specific Questions*
*1.Understand the Document Summary*

- *Thoroughly read and understand the content of the document for which you are creating questions.*
- *Identify key facts, events, and figures that are central to the document's narrative.*

*2. Contextual Relevance*

- *Ensure each question is directly related to the content of the document.*
- *Provide enough context within the question to uniquely identify the relevant document from a set of similar documents.*

*3. Specificity in Question Framing*

- *Frame your questions to be specific and direct.*
- *Use proper nouns and specific details instead of general terms. For instance, ask "Where was John Smith born?" rather than "Where was he born?"*

*4. Avoid Vague Language*

- *Eliminate vague phrasing and general inquiries that could apply to multiple documents.*
- *Each question should point clearly to a unique answer within a specific document.*

*5. Clarity and Precision*

- *Questions should be clearly phrased to avoid ambiguity.*
- *Ensure that the language used does not leave room for multiple interpretations.*

*6. Incorporate Critical Details*

- *Include critical details in the question that are necessary to guide the user to the right document.*
- *For example, if discussing events, specify the event's date or location if these details are crucial for identifying the correct document.*

*7. Test Questions for Specificity*

- *After drafting a question, check if it can be answered by more than one document.*
- *Revise the question to include more specific details if it is not unique enough.*

*8. Feedback and Revision*

- *Seek feedback on the questions from peers or supervisors to ensure they meet the specificity and clarity criteria.*
- *Revise questions based on feedback to enhance precision and contextual relevance.*

*9. Consistency Check*

- *Regularly review your questions for consistency in style and the level of detail required.*
- *Ensure that all questions adhere to the project's guidelines and standards.*

---

**Vendor Quote 3**

*Answer Annotator:*

*1. Answer Based on Document*

- *Ensure that your answer is directly based on the information provided in the document linked to the question.*
- *Do not assume or use external knowledge that isn't found in the document.*

*2. Quality Over Length*

- *Focus on the quality, accuracy, and relevance of the answer rather than its length.*
- *Answers can be brief or extended as needed, as long as they fully address the question.*

*3. Rephrasing for Clarity*

- *If the information in the document is not in a complete sentence, rephrase it into one.*
- *Ensure the answer is complete, grammatically correct, and standalone.*

*4. Direct and Relevant Answers*

- *Start the answer with the most direct piece of information (the "actual answer") as early as possible in the response.*
- *Follow up with details or clarifications as necessary.*
- *If you cannot find the answer in the document, simply mention 'the answer is not found in 'the document.'*

*5. Yes/No and Specific Answers*

- *If applicable, start the answer with "Yes" or "No".*
- *For questions that require specific data (like dates, numbers, or names), begin with that specific answer before elaborating.*

*6. Conciseness*

- *Provide concise answers without unnecessary filler. If a short answer suffices, use it.*
- *Aim for directness and utility in every response.*

*7. Long Answer Annotation*

- *Clearly mark the paragraphs in the document from which the answer was derived (answer pointer).*
- *If the document's text directly answers the question, it should be copied under the "long answer" section to provide context.*

*8. Verification and Revision*

- *Double-check the answers against the document to ensure they are accurate and complete.*
- *Revise answers based on feedback from peers or supervisors to enhance accuracy and clarity.*

*9. Consistency and Formatting*

- *Maintain consistent formatting in how answers and supporting information are presented.*
- *Ensure that all parts of the instructions are followed for each answer to maintain a uniform standard across the project.*

---

We asked the Vendor to confirm whether the above were the full extent of instructions provided to the Content Creators and Annotators. Their replies is given in Vendor Quote 4.

---

**Vendor Quote 4**

*The provided enumerations for Content Creators, Question Annotators, and Answer Annotators are indeed the full extent of the instructions handed to the participants. These comprehensive instruction sets were designed to cover all aspects of their respective tasks in detail.*

*In addition to these written guidelines, participants received further training through online sessions conducted via Zoom. These sessions also included calls to address any questions or issues, ensuring that all participants thoroughly understood their responsibilities and the procedures to follow.*

---

Initially, the Vendor didn't provide us with similar instructions for Quality Control Annotators. We explicitly asked for it, and they provided Vendor Quote 5.

---

**Vendor Quote 5**

*Quality Check Specialist Instruction Set for Content Creation, Question Annotation, and Answer Annotation*

*For Content Creation:*
*1. Compliance with Guidelines: Verify that all created documents strictly adhere to the project's guidelines on format, style, and themes.*
*2. Originality Check: Use plagiarism detection tools to ensure all documents are original and free from copied material.*
*3. Content Quality Review: Assess the clarity, engagement, and grammatical correctness of the content. Use MS Word grammar and spell-check tools.*
*4. Diversity and Inclusivity Audit: Ensure that the content reflects diversity in characters, settings, and scenarios and adheres to inclusive practices.*
*5. Legal and Ethical Compliance: Confirm that the content does not violate any legal or ethical standards and is free from offensive or controversial material.*

*For Question Annotation:*
*1. Ensure that each question directly relates to the document and includes specific details that tie it uniquely to that document.*
*2. Review questions for clarity to avoid ambiguity, ensuring they are straightforward and lead to a singular, clear answer.*
*3. Verify that all questions maintain a consistent format and level of detail, adhering to the project's guidelines.*

*For Answer Annotation:*
*1. Check that each answer is accurately based on the content of the specific linked document and not inferred from external knowledge.*
*2. Ensure answers are complete and provide all necessary information to fully address the question. Avoid leaving out crucial details.*
*3. Confirm that answers start with the most direct response and are structured to prioritize the primary information first.*
*4. Review answers for unnecessary verbosity. Ensure answers are concise yet thorough.*
*5. Maintain a uniform standard in presenting answers and supporting information, adhering to the project's formatting guidelines.*

---

## D.2 Additional Information

In complement to the instructions listed in Appendix 1, this section contains additional information provided by the Vendor during our exchanges. While there is some overlap with these instructions, we

presume that the additional information could have been conveyed during the Zoom online sessions mentioned in Vendor Quote 4. Note that some of our questions were asked before the instructions listed in Appendix 1 were provided.

For a given document topic among the 17 possibilities, we asked how the Content Creators proceeded to further refine the general content of the document: what is the creative process?

---

**Vendor Quote 6**

*For creating documents within the specified [topic] for the dataset project, the process of deciding on the general topic involved a structured approach to ensure diversity and relevance:*

*a. Selection of Document [topic]: The initial step was selecting a document [topic] from the provided list, such as Local News, SMEs, or Local Health and Wellness. This choice was driven by the current focus of the project needs and the demand for diverse types of documents in our data annotation tasks.*

*b. Subtopic Identification: Within each chosen [topic], we used to identify specific subtopics. For instance, if the [topic] was Local News, subtopics might include City Council Decisions or Community Events.*

*c. Current Trends and Relevance: We used to select topics based on their current relevance or trends. For instance, in Local Health and Wellness, recent public health campaigns or new wellness workshops were chosen as topics to ensure the dataset is up-to-date and useful for contemporary models.*

*d. Innovation and Imagination: For synthetic documents, like Imaginary Company Policies or Synthetic News Stories, creativity plays a significant role. Here, topics were created by imagining scenarios or policies that are plausible yet innovative, ensuring that these documents are unique.*

---

We asked if Content Creators had prior knowledge of the topics they wrote on.

---

**Vendor Quote 7**

*Yes, the team had prior knowledge of the given topic. Our team continually updates their knowledge based on current trends and developments. The team also included experienced content creators who are skilled in writing synthetic documents that mimic real-world scenarios and documents, which was essential for categories that require imaginative yet plausible content.*

---

We asked if the Content Creators actively conducted research on the topic, and if yes what was the nature of this research.

---

**Vendor Quote 8**

*Yes, the team conducted comprehensive research to support the creation of the diverse document dataset. The research was multifaceted, aiming to ensure the documents were both informative and compliant with the requirements. For each document category and subtopic, the team analyzed current trends and developments to choose topics that are timely and relevant. This included monitoring recent events, emerging issues, and ongoing discussions within specific domains such as local politics, technology, and health.*

---

We asked if online resources or documents were used to gather information before writing.

---

**Vendor Quote 9**

*Yes, the team utilized a variety of online resources and documents to gather necessary information before creating the dataset. This preparatory step was crucial to ensure that the content was both accurate and comprehensive.The team accessed several academic databases and digital libraries to retrieve relevant research papers, articles, and case studies. This was particularly important for understanding current trends and methodologies in the areas of interest, such as cybersecurity, local governance, and public health. For up-to-date information and to ensure the relevance of the topics selected, the team regularly consulted industry-specific reports. News websites, especially those covering local news and developments, were also a critical resource for identifying recent events and changes that could influence document topics. To gain deeper insights into niche topics such as local arts scenes or regional folklore, the team engaged with specialized forums and online communities. These platforms often provide firsthand accounts and discussions that are not available through mainstream sources.*

---

As mentioned in Appendix 1, the Vendor later clarified that "No language model was used by the content writers while creating the content for the synthetic documents."

We asked what was the process for deciding on various names of places, people, and organizations within the texts.

---

**Vendor Quote 10**

*For the dataset involving synthetic documents and content, the process for deciding on names of places, people, and organizations was carefully managed to ensure realism while avoiding the use of real-world entities that could lead to legal or ethical issues.*
*Generating Names:*
***Random Name Generators***: *For generating plausible yet fictitious names for people, places, and organizations, the team utilized random name generators. These tools can be adjusted to create names that are culturally or regionally appropriate, adding to the realism of the synthetic documents.*
***Creative Invention***: *For certain documents, especially those requiring a unique or thematic naming convention (like a company involved in green technologies or a fictional local government initiative), names were creatively invented by the content team. This allowed for a controlled integration of the names into the context of the documents.*
***Avoiding Real Names***: *Names generated or invented were cross-referenced against existing entities using online searches and databases to ensure they did not inadvertently correspond to real people, places, or organizations. This step was crucial to prevent potential legal issues or unintended associations.* ***Modifying Common Names***: *In some cases, common names were slightly altered to create a unique but believable alternative. This practice helped in maintaining the authenticity of the text while ensuring the names were fictitious.*
***Thematic Consistency***: *In scenarios or documents themed around specific industries or societal issues, names were selected or created to resonate with the theme. For example, a tech startup was named with a modern, innovative twist.*

---

Asked about the quality control process, they replied the following.

---

**Vendor Quote 11**

*Quality Control Process*

- ***Peer Review***: *Each document, along with its associated questions and answers, was reviewed by a second team member to check for accuracy, relevance, and adherence to guidelines.*

- ***Expert Review:*** *Subject matter experts reviewed samples to ensure that the content was plausible and well-aligned with the intended scenarios or topics.*
- ***Automated Checks:*** *Software tools were used to check for grammatical errors, plagiarism (to ensure originality of synthetic documents), and formatting consistency.*

*[Cleanup Process]*

- ***Corrections:*** *Any issues identified during the QC checks were corrected, which included revising content, reformatting documents.*
- ***Final Approval:*** *Documents underwent a final review by the QC lead before being deemed ready for inclusion in the dataset.*

---

We asked for confirmation whether only the summary, not the reference documents, were provided to the Question Annotators.

---

**Vendor Quote 12**
*The question and annotation task was done on our platform by the annotators. On our platform, the users were shown dynamically generated summaries of the documents to create questions. Once they created their questions, the users answering the questions were provided with the whole document to formulate their answers. Since these were generated dynamically, we may not be able to retrieve the exact same summaries which were provided to annotators.*

---

With the understanding that there is no guarantee that it provides the exact same summaries as those provided to the Question Annotators, we asked for a high-level description of the technology used to generate those summaries, and for an example of such summaries for the documents listed in Eq. (1).

---

**Vendor Quote 13**
*Process Description*

- *Extracting Text from PDF Documents:*
  *PDF documents were converted to DOCX format to facilitate text extraction. The python-docx library was utilized to read the DOCX files and extract the text content. The extracted text was then processed paragraph by paragraph. Each paragraph was summarized using the spaCy natural language processing library. The summarization involved reducing the length of each paragraph while preserving the main points.*
- *Generating Summarized PDFs:*
  *The summarized text was compiled and formatted. The FPDF library was used to create new PDF files containing the summarized content.*
- *Technologies and Libraries Used*
  *Python: The primary programming language used for the entire process.*
  *SpaCy: A powerful natural language processing library used for summarizing the text. The en_core_web_sm model was loaded to process the text and extract sentences for summarization.*
  *python-docx: A library for creating, modifying, and extracting text from DOCX files. FPDF: A library for creating PDF documents.*

---

The Vendor provided us with the 10 examples we requested. The most summarized of these documents has 57% as many words as the corresponding original, the least summarized had 69%, and the mean value for these 10 documents is 65%.

# E  Related Work

**Question-Answering Datasets.**  The domain of question answering (QA) in natural language processing has seen the development of numerous datasets that serve as the benchmark for evaluating the capabilities of various language generation models [Rajpurkar et al., 2016, 2018, Joshi et al., 2017, Kwiatkowski et al., 2019, Dasigi et al., 2019]. Datasets such as SQuAD, SQuAD2.0 [Rajpurkar et al., 2016, 2018] and HotpotQA [Yang et al., 2018] primarily consist of single-span instances where the answers are confined to a single text span from the provided context. Later datasets such as TriviaQA [Joshi et al., 2017], and QuAC [Choi et al., 2018] have been developed to address this bias issues.  In addition, DROP [Dua et al., 2019] which requires arithmetic reasoning, and the HAS-QA [Pang et al., 2019], specializing in healthcare, further expand the complexity and range of real-life QA tasks. However, the majority of these datasets typically derive their content from high-traffic, publicly accessible websites [Chen et al., 2017], and can possess a significant risk of data contamination whereby models may recall rather than understand and respond to the question [Shi et al., 2024] Secondly, the reliance on extractive answers or specific answer formats can limit the ability of models to generate novel, contextually rich responses. In contrast, RepliQA aims to overcome these limitations by ensuring no data contamination, offering a larger and more diverse dataset, and employing a controlled release process to prevent premature exposure and maintain the integrity of the evaluation.

**Large Language Models.**  LLMs have revolutionized the language generation capabilities and are typically trained on extensive datasets compiled from the internet [Devlin et al., 2018, Peters et al., 2018, Wang et al., 2021, Dettmers et al., 2022, Touvron et al., 2023, Jiang et al., 2023]. BERT [Devlin et al., 2018] primarily trained on bookcorpus and wikipedia, while T5 [Wang et al., 2021, 2023] leverages the C4 dataset, and GPT variants utilize diverse range of internet text for pre-training. The fact that the training sets used to trained LLMs contains text also included in benchmarks is a known issue when evaluating LLMs [Dodge et al., 2021, Xu et al., 2024b].  This overlap can occur if a dataset is publicly accessible and inadvertently included in the data crawled during model training, or if it is derived from sources that also contribute to training corpora [Dodge et al., 2021]. Additionally, when assessing closed-source LLMs via interfaces, there is no assurance that the developers have not utilized insights from previous benchmarking initiatives to enhance model performance [Balloccu et al., 2024].  This underscores the need for unseen datasets in the evaluation of LLMs to ensure genuine performance assessments.

**Topic Retrieval.**  Our work also intersects with in-context learning works that evaluate LLMs on text classification. LLMs when they scale to billions of parameters, have shown emerging adaptability properties across a spectrum of tasks, notably through instruction tuning and in-context learning (ICL) [Brown et al., 2020, Ouyang et al., 2022, Chowdhery et al., 2023].  This approach enables them to learn the patterns illustrated by the demonstrations, allowing to solve new tasks during test time effectively without any fine-tuning. This ability has been useful not only in generative tasks like question answering or summarization but also promising for discriminative tasks such as text classification [Milios et al., 2023, Lu et al., 2022, Wei et al., 2021, Liu et al., 2022, Edwards and Camacho-Collados, 2024]. Furthermore, to enhance the efficiency of ICL, recent advancements have integrated Retrieval-Augmented Generation (RAG) [Lewis et al., 2020]. This integration improves the performance and adaptability of LLMs across these tasks by enabling them to access and utilize relevant external knowledge dynamically [Ram et al., 2023].

**Data Leakage.**  As the size of pre-training datasets for LLMs continues to grow, the risk of inadvertently incorporating evaluation data into these training corpora increases. Furthermore, with the trend towards less transparent training processes, instances of data leakage becomes increasingly common [Shi et al., 2024]. Several studies have highlighted these concerns, indicating that traditional evaluations and benchmarks may not effectively challenge or accurately reveal the true capabilities of these models[Aiyappa et al., 2023, Zhou et al., 2023, Balloccu et al., 2024, Dodge et al., 2021, Oren et al., 2024, Zhang et al., 2024]. This underscores the importance of developing datasets like RepliQA, which are designed to be unseen by LLMs, thus providing a true test of their learning and generalization capabilities rather than mere recall.

# F  Prompts and other inference details

We set inference paratemers to defaults defined by OPENROUTER[7]. Prompts for each evaluation are reported below.

---

**Question-Answer Prompt**

*System prompt*: You are an expert research assistant, skilled in answering questions concisely and precisely, using information provided by the user.

*User prompt*: I'd like for you to answer questions about a context text that will be provided. I'll give you a pair with the form:

Context: "context text"
Question: "a question about the context".

First, tell me about your knowledge of the context and what information it contains, then, create an analysis of the context strictly using information contained in the text provided. Your knowledge about the context and the analysis must not be output. Finally, generate an explicit answer to the question that will be output. Make sure that the answer is the only output you provide, and the analysis of the context should be kept to yourself. Answer directly and do not prefix the answer with anything such as "Answer:" nor "The answer is:". The answer has to be the only output you explicitly provide. The answer has to be as short, direct, and concise as possible. If the answer to the question can not be obtained from the provided context paragraph, output "UNANSWERABLE". Here's the context and question for you to reason about and answer:

Context: {context}
Question: {question}

---

**No-Context Question-Answer Prompt**

*System prompt*: You are a helpful assistant and an expert all knowing genius, well versed in trivia about all sorts of topics and able to respond concisely and precisely about any question a user may ask.

*User prompt*: I'd like for you to answer questions about a topic I'm interested about. I'll provide you with a question of the form:

Question: "a question".

First, tell me about your knowledge of the topic from which the question stems, then, create an analysis of the topic using your knowledge about it. However, your knowledge about the topic and the analysis must not be output. Finally, generate an explicit answer to the question that will be output. Make sure that the answer is the only output you provide, and the analysis of the topic should be kept to yourself. Answer directly and do not prefix the answer with anything such as "Answer:" nor "The answer is:". The answer has to be the only output you explicitly provide. The answer has to be as short, direct, and concise as possible. If the answer to the question can not be obtained from your existing knowledge, output "UNANSWERABLE". Here's the question for you to reason about and answer:

Question: {question}

---

[7] https://openrouter.ai/

> **Topic retrieval Prompt**
>
> *System prompt*: You are an expert research assistant, skilled in answering questions concisely and precisely, using information provided by the user.
>
> *User prompt*: "I'd like for you to determine the topic of some text context that will be provided. I'll give you the text with the form:
>
> Context: "text".
>
> First, tell me about your knowledge of the context and what information it contains, then, create an analysis of the context strictly using information contained in the text provided. Your knowledge about the context and the analysis must not be output. Finally, generate an explicit topic of the context by choosing from the following list of topics:
>
> {topics}
>
> Make sure that the topic is the only output you provide, and the analysis of the context should be kept to yourself. Answer directly with the topic from the list, and do not prefix the answer with anything such as "Answer:" nor "Topic:". The topic has to be the only output you explicitly provide. Your output has to be as short, direct, and concise as possible. The answer strictly has to be one of the topics provided to you. If the topic cannot be obtained from the provided context paragraph, output "UNANSWERABLE". Here's the context for you to determine the topic:
>
> Context: {context}

## G   Topic classification results

We tested to what extent topics in REPLIQA are well defined in the sense that there's little confusion on a topic given a document. To do so, we trained a topic classifier given by a pre-trained LONGFORMER encoder [Beltagy et al., 2020]. Trained classifiers achieved test accuracy greater than 95%, which suggests that overlap is not a reason for concern. Code implementing such experiment can be found at: `https://github.com/ServiceNow/repliqa/blob/main/repliqa_topic_classifier.ipynb`. In Figure 12, we report a confusion matrix showing little topic ambiguity.

## H   Evaluating models fine-tuned for question-answer tasks

While our main evaluations focused on general purpose language models, we additionally evaluated a model that was fine-tuned specifically for question-answering tasks. Namely, we evaluated LLAMA3-CHATQA-1.5-8B (indicated as fine-Tuned in the bottom row of table 8). We observed a performance degradation on REPLIQA and a slight improvement on TRIVIAQA, which is in alignment with evaluations of other fine-tuned models on a pre-release version of REPLIQA and other datasets, as reported in [Monteiro et al., 2024]. Note that we closely followed the prompt format recommended in `https://huggingface.co/nvidia/Llama3-ChatQA-1.5-8B`. Inference code can be found at: `https://github.com/ServiceNow/repliqa/blob/main/hf_qa_eval.ipynb`.

## I   Length distributions of REPLIQA and TRIVIAQA

Figures 13 and 14 show length histograms for the context documents REPLIQA and TRIVIAQA, respectively[8]. Note that TRIVIAQA's documents are generally much longer than REPLIQA's, and one would expect that to yield better performance on REPLIQA since shorter documents will make it easier for answers to be found.

---

[8]The rightmost histograms in both figures corresponds to tokenization with the following tokenizer: `https://huggingface.co/google-bert/bert-base-uncased`.

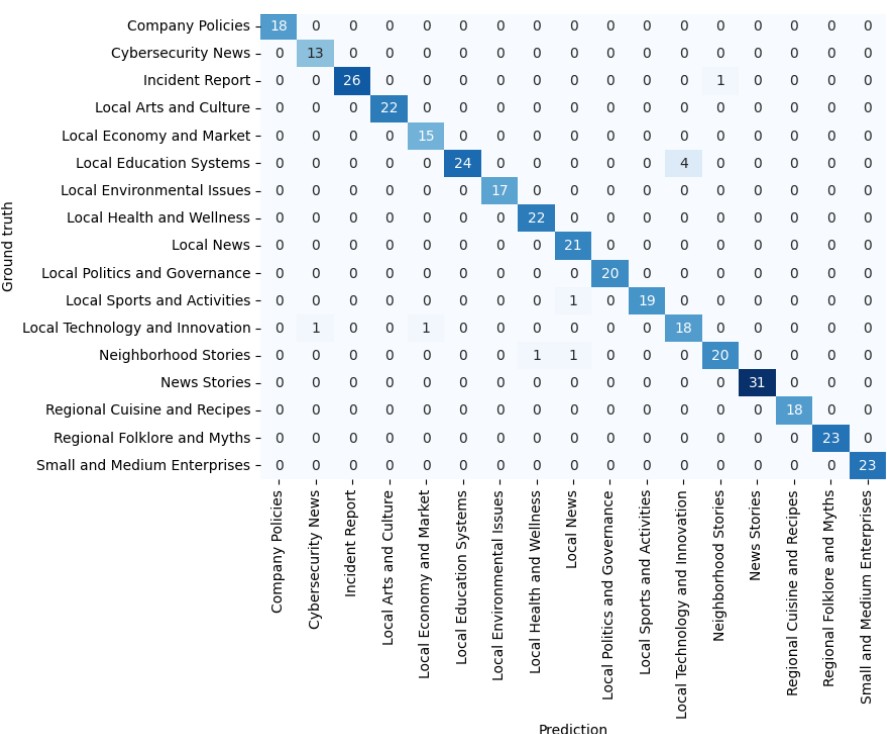

**Figure 12:** Confusion matrix for predictions of a topic classifier trained on REPLIQA.

**Table 8:** Question-Answer performance of a fine-tuned model in terms of recall.

|  | REPLIQA | TRIVIAQA |
|---|---|---|
| *Claude 3 Sonnet* | 66.54 | 75.41 |
| *Mistral Large* | 72.29 | 86.79 |
| *LLama 3 8B* | 54.82 | 70.95 |
| *LLama 3 8B (Fine Tuned)* | 57.19 | 63.51 |

This is in contrast with results in our benchmarking where models consistently perform much better on TRIVIAQA, which serves as further evidence that models mostly know the answers to TRIVIAQA's questions and tend to ignore input context documents. This effect happens to an extent that documents being longer in TRIVIAQA will not impact performance negatively.

A notebook implementing the plots of the histograms is made available at: `https://github.com/ServiceNow/repliqa/blob/main/length_distributions.ipynb`.

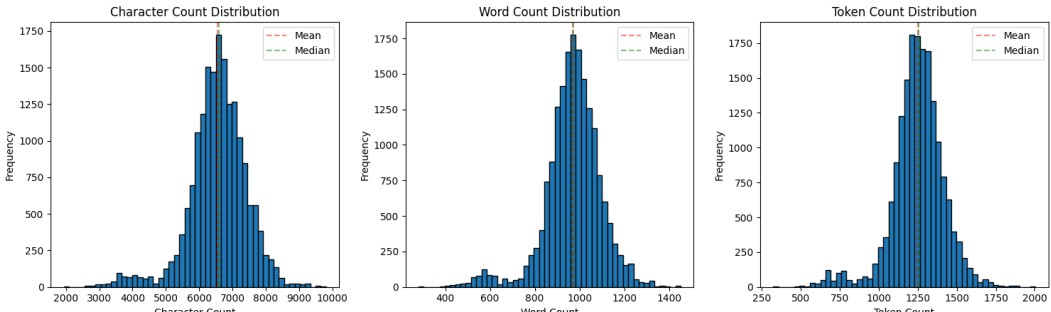

**Figure 13:** Length histograms: REPLIQA

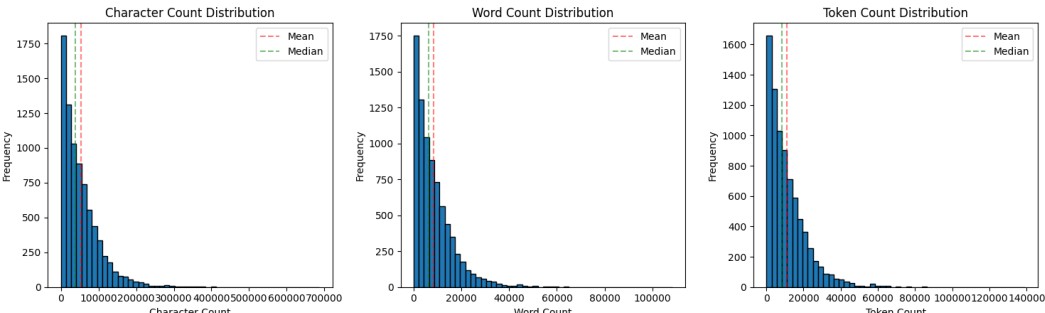

**Figure 14:** Length histograms: TRIVIAQA

