# OpenReview forum: "RepLiQA: A Question-Answering Dataset for Benchmarking LLMs on Unseen Reference Content"
_NeurIPS.cc/2024/Datasets_and_Benchmarks_Track — NeurIPS 2024 Track Datasets and Benchmarks Poster_

### Official Review · Reviewer_a4ei · 2024-06-24

**Rating:** 7
**Confidence:** 4
**Correctness:** Claims appear to be sound.
**Clarity:** The paper is clear.

**Review:**

This dataset represents, I believe, a substantial contribution to the community. It is unclear whether the license allows for commercial use or not: CC-by-4.0 on [HF](https://huggingface.co/datasets/ServiceNow/repliqa) vs. CC-by-NC-4.0 in the paper. I think that the former would be more useful if the authors want their dataset to be more widely used in industry. Although it's not entirely clear how extensively LLMs were used in creating this dataset (the results in Fig. 9 suggest some involvement), the authors made a thorough effort to ensure data quality. One potential issue is the annotators' pool: all annotators were based in India and presumably wrote all the documents in the dataset. Their writing style may differ from that of other English speakers, yet this is not discussed in the paper. The release plan to mitigate data leakage is a smart strategy, although it would have been helpful for reviewers to have access to the data for a more comprehensive evaluation. Lastly, the paper is well written and clear.

**Strengths:**

Detailed above.

**Additional Feedback:**

No additional feedback.

**Documentation:**

Data collection process has been documented.

**Ethics:**

No concerns.

**Limitations:**

Mentioned above. More discussion of the limitations is needed.

**Opportunities For Improvement:**

Miscellaneous questions and concerns:
- Can the authors discuss the limitations of the annotators' pool, specifically addressing potential biases in content creation, consistency and quality control, limited exposure to diverse topics, and differences in language and grammar conventions between Indian English and other variants of English? In particular, the dataset will not reflect the language of non-native speakers or even underrepresented communities within English-speaking countries. This must be discussed in the paper.
- The results in Figure 9a suggest that LLMs were used in creating this dataset. In the HF repo, the authors claim: “Scoring RepLiQA documents with Fast-DetectGPT results in scores that are notably different from those of FineWeb.” How do the findings in the image support this statement?
- In Figure 5, why does GPT-3.5 Turbo have fairly high recall without context on RepliQA?
- The finding that smaller models beat larger models is surprising and needs further investigation. Are the authors sure that this cannot be attributed to data quality? I’m not convinced by the explanation that any improvement is due to memorization: We have lots of other datapoints that indicate that larger models are simply better at performing this type of task.
- Please improve Figure 4. Currently, it’s hard to identify the same model across facets. Consider plotting the bars horizontally or the datasets side by side for each model.
- Why is the finding in 4.3 interesting? Because the context document is fictitious and thus there is no actual answer unless the document is provided?
- Are the splits all randomly sampled documents from the entire pool? What should we expect from the next shards?

**Relation To Prior Work:**

Prior work is discussed.

**Summary And Contributions:**

The paper introduces RepLiQA, a novel dataset created to evaluate the performance of large language models (LLMs) on question-answering and topic retrieval tasks using unseen reference content. The dataset consists of five splits, four of which have not been released yet. Each sample includes a reference document crafted by a human annotator, a question about the document, a ground-truth answer, and the paragraph containing the answer. This ensures models must read and understand the provided document to generate accurate answers, minimizing the impact of pre-trained knowledge. The authors assess the performance of 18 LLMs on these tasks, providing insights into their ability to rely on  internal memory versus reading the provided documents. They also benchmark the same set of LLMs on TriviaQA and, based on the differences in the results, argue that some models rely heavily on memorization rather than actual reading comprehension.

---

> ### Author Rebuttal · Authors · 2024-08-16
>
> We sincerely thank the reviewer for their detailed feedback. We are glad the reviewer found that RepLiQA represents a substantial contribution to the community, that the release plan to mitigate data leakage is a smart strategy, and that the paper is well written and clear.
>
>   - **CC-by-4.0 on HF vs. CC-by-NC-4.0 in the paper.**
>
> Thank you for pointing out this mistake. Please note that we release under a CC-BY-4.0 license as stated in the HF repo. The paper was fixed.
>
>   - **Can the authors discuss the limitations of the annotators' pool**
>
> Thank you for this question. We inherit the risks of any annotation task of such a large scale and use human and automated quality controls to mitigate the risks to a satisfactory extent.
>
> Our Limitations section has been amended to reflect the possible bias toward Indian English with respect to other variants of English, and in particular of non-Indian non-native English speakers. That being said, this kind of bias (if present) would be more important in a dataset meant for training, which RepLiQA is not. RepLiQA is meant for benchmarking a model’s ability to answer (or identify as unanswerable) questions based on a never-seen-before context. A priori, this capability should be orthogonal to the model’s performances in different English dialects.
>
> Moreover, our manual inspection of the dataset indicates that a rather “international” English dialect is used. We respectfully invite the reviewer to visually inspect a few samples in RepLiQA just in case they didn’t have a chance to do so yet since that would help get a sense of the level of quality we achieved. This can be done somewhat easily and directly on the browser at https://huggingface.co/datasets/ServiceNow/repliqa.
>
>   - **In the HF repo, the authors claim: “Scoring RepLiQA documents with Fast-DetectGPT results in scores that are notably different from those of FineWeb.” How do the findings in the image support this statement?**
>
> Results in figure 9-a show that detection scores for RepLiQA’s content are non identically distributed relative to Alpaca (known to be synthetic) and FineWeb (known to be natural). While this observation doesn’t yield any clear conclusion, we cannot discard the possibility that there were LLM interventions during the annotation, although we highlight once more that we could not find strong evidence for that being the case.
>
> We highlight however that, as discussed in our Limitations section, analysis is performed to help understand the process of data creation, and the only requirement RepLiQA is expected to satisfy is for it to represent non-factual but natural looking data. The quality controls, the post-annotation human/automated inspections, and the QA results we reported all suggest those requirements are satisfied to a great extent.
>
>   - **In Figure 5, why does GPT-3.5 Turbo have fairly high recall without context on RepliQA?**
>
> This is due to limitations of the performance metric. We refer the reviewer to Table 6 where we report other metrics and show that the same model scores 0 in terms of exact match as one would expect. More verbose models tend to gain recall points due to eventually outputting parts of an answer correctly. Note that replacing recall for some other metric would introduce other issues, and we chose recall for consistency with previous literature since it is widely used in QA settings.
>
> We highlight that even if the metric is not perfect (as in it is not 0 for a model that only answers wrongly), it does show evidence to support our claim: models can answer TriviaQA’s questions without context to a great extent, and do not improve much once contexts are given, suggesting they rely on memory to a great extent. The behaviour on RepLiQA is also as expected and models fail consistently without contexts, serving as a sanity check and showing evidence that the data we release is indeed such that reference documents are required to answer questions accurately.
>
>   - **The finding that smaller models beat larger models is surprising and needs further investigation.**
>
> We would like to respectfully dispute the surprisingness of that finding. Larger models have been observed to perform better *on average* across tasks and benchmarks, but not for every task across every dataset. In almost all cases on TriviaQA, models behaved as expected and bigger models were better than smaller ones.
>
> On RepLiQA on the other hand, the situation is rather mixed and results not consistent at all since we see both behaviours manifest. We hypothesize that that is due to larger models being more prone to rely on memory, but confirming that is the object of future work. We also intend to investigate what happens in  situations where the user-provided context contradicts model memory for instance, in which case we expect larger models to be more strongly affected by contradictions and more likely to ignore given contexts and rely on training knowledge, as suggested by our results.
>
>   - **improve Figure 4. Currently, it’s hard to identify the same model across facets.**
>
> Thank you for the suggestions. The attached pdf contains a new plot following the suggestion of the reviewer, which we included in the manuscript to ensure readability of those results.
>
> Kindly note that the bottom left plot in the current version of figure 4 contains the gaps for each model across the two datasets, and a complete set of results is also reported in table 3 within Appendix A covering more metrics.
>
> **We provide further responses in the comment below.**

---

> > ### Author Response · Authors · 2024-08-16
> >
> > - **Why is the finding in 4.3 interesting?**
> >
> > We remark that questions posed to models in that evaluation cannot be answered, with or without reference documents, so models are expected to refuse answering 100% of the time. We originally hypothesized that models that were given a context would be more prone to hallucinating answers that do not exist in the document but the experiments reported in 4.3 actually showed the opposite. Models that are given a context, even if it doesn’t contain the answer, refuse to answer more often. Models prompted without contexts on the other hand tend to hallucinate answers quite often. In practice, this finding suggests that settings such as retrieval augmented generation have an extra benefit of yielding models that refuse more reliably.
> >
> >   - **Are the splits all randomly sampled documents from the entire pool? What should we expect from the next shards?**
> >
> > Yes, we made sure to split the data in such a way that only new documents appear in new splits. It is not as simple as “all randomly sampled documents from the entire pool”, but the ultimate result is basically the same.
> >
> > To be extra punctilious, 10 samples (listed in Appendix B’s Equation (1)) were manually assigned to RepLiQA_0 (because they were used in online APIs as early as January 2024). The remaining samples were randomly sampled without replacement and each assigned to one of the five splits so that the 17 different topics are evenly distributed. All splits are thus independently distributed by construction, and each document appears in exactly one split.
> >
> > Training on RepLiQA_0 today may confer a minor advantage on RepLiQA_1 performance, but that advantage would be solely due to the model learning the “style” of RepLiQA’s documents, questions and answers: no “fake facts” would leak. Moreover, models that were never trained on any RepLiQA splits should perform similarly well on all RepLiQA splits. This means that if a model performs significantly better on an “old” RepLiQA split than on new ones, that model was likely trained on the old split. Our staggered release schedule may thus be seen as an experiment enabling the study of interesting questions. Which models trained on RepLiQA? At what rate does such leakage occur?
> >
> > We also remark that this sequential release approach enables improving the dataset over time since we have the opportunity to incorporate the community’s feedback or fix any issues identified in earlier releases. That is, while we diligently (and arguably excessively) controlled for quality, any community feedback for something that still escaped our controls will be dealt with before the next releases.

---

> > > ### Comment · Reviewer_a4ei · 2024-08-19
> > >
> > > Thanks for your clarifications. I have updated my score.

---

> > > > ### Author Response · Authors · 2024-08-23
> > > >
> > > > Thank you for your engagement with the discussion.

---

### Official Review · Reviewer_GtR5 · 2024-07-14

**Rating:** 7
**Confidence:** 3
**Correctness:** yes
**Clarity:** yes

**Review:**

# Strengths:

1. Addressing Dataset Contamination: The development of REPLIQA effectively tackles a crucial challenge in LLM evaluation, which is the potential contamination of test datasets. By ensuring that the reference documents are completely new and unseen, the authors provide a more robust and accurate benchmark for evaluating LLM performance.

2. Extensive Experiments: The paper presents a thorough evaluation involving 18 state-of-the-art LLMs, providing substantial insights into their performance on previously unseen data. This extensive benchmarking aids in a comprehensive understanding of the strengths and weaknesses of different models, thereby contributing significantly to the field.

3. High-Quality, Human-Curated Content: The dataset's reliance on human annotators for generating reference documents and question-answer pairs ensures that the content is of high quality and contextually relevant. This human curation guarantees that models are evaluated on their ability to accurately interpret and utilize the provided information.

# Weaknesses:
1. Formatting Issue: It appears that the paper uses the camera-ready version instead of the submission version. It is recommended to follow the correct submission template for NeurIPS 2024.

2. Details on Annotator Agreements and Data Quality: The paper could benefit from including more detailed information about the agreements with different annotators and the measures taken to ensure data quality. This additional information would provide greater transparency and strengthen the credibility of the dataset creation process.

3. Topic Retrieval Ambiguity: In the topic retrieval task, there is a possibility that a single document might belong to multiple categories. Addressing this potential ambiguity and providing clear guidelines on how such cases are handled would enhance the clarity and robustness of the evaluation.

4. Consideration of other Fine-Tuned LLMs: It would be advantageous to consider including fine-tuned LLMs tailored specifically for QA tasks, such as ChatQA (available at https://huggingface.co/nvidia/Llama3-ChatQA-1.5-8B, https://huggingface.co/nvidia/Llama3-ChatQA-1.5-70B). This addition could provide valuable insights into the performance of models that have been specifically optimized for question-answering tasks.

**Strengths:**

See above.

**Additional Feedback:**

n/a

**Documentation:**

yes

**Opportunities For Improvement:**

See above.

**Relation To Prior Work:**

yes

**Summary And Contributions:**

The paper introduces REPLIQA, a novel dataset designed to benchmark Large Language Models (LLMs) on unseen reference content. REPLIQA contains approximately 90,000 question-answer pairs and 18,000 reference documents, carefully curated to prevent data contamination from existing internet sources. The primary aim is to assess the ability of LLMs to generate accurate answers and identify document topics based on provided contexts, without relying on pre-trained knowledge.

---

> ### Author Rebuttal · Authors · 2024-08-16
>
> Thank you for your useful and actionable feedback. Thank you also for highlighting that RepLiQA consists of high-quality human-curated content that effectively tackles a crucial challenge in LLM evaluation, and that our evaluations are thorough.
>
>   - **the paper uses the camera-ready version instead of the submission version.**
>
> We highlight that we used the 2024 official template for the datasets and benchmarks track, which is slightly different relative to the one for the main track. We did choose the option to not anonymize the submitted version, which is allowed according to the call for papers for this track. Among other things, this non-anonymity choice simplifies the distribution of the dataset through HuggingFace to both anonymous referees and to the general community.
>
>   - **Details on Annotator Agreements and Data Quality**
>
> We kindly refer the reviewer to Appendix B where we highlight issues we personally identified and rectified. Also, in Appendix C.1 we transparently disclose further potential irregularities we observed but left unaltered: less than 1% of the dataset is concerned and these irregularities were small issues or left behind noise from the annotation process. We believe that releasing these samples verbatim will help the community better understand the dataset in the future. Finally, Appendix D shows details concerning the annotation process and the instructions given to annotators, including the various quality checks performed by the vendor and annotators.
>
> We additionally mention that the released dataset is in a hugging face repository (https://huggingface.co/datasets/ServiceNow/repliqa), which has a very useful data preview tool that enables easy visualization of the data directly on the browser, so that quality issues can be easily found if they exist. The page also supports issue reporting, and any reported quality concern will be treated by the repo’s maintainers and confirmed quality problems will be removed, if any was left behind after all the filters applied during the development phase.
>
>   - **In the topic retrieval task, there is a possibility that a single document might belong to multiple categories.**
>
> Thank you for this comment since this is of high importance for future releases. Indeed, topics can be ambiguous although the strong results we already observed in the manuscript (i.e., F1 scores above 70%) suggest that’s not a major issue, which is in agreement with what was observed during quality controls.
>
> To further verify that topics are well defined, we add a new experiment in which we trained a topic classifier on top of RepLiQA and easily achieved test accuracy greater than 95% with a small BERT, which further supports the claim that topic overlap is not a major issue. For the sake of full transparency, a simple colab notebook showing those results can be found here: https://colab.research.google.com/drive/1l6TgMzlo3X9Ev4bfhsvN8-eA8MkBeMk-?usp=sharing . We also attached a confusion matrix to this rebuttal, and these experiments are now added to the manuscript.
>
> That being said, because the dataset is public, the community could identify issues such as ambiguities between different classes. In such cases, our staggered release schedule allows us to address those issues in future splits (provided that we can confirm the issues ourselves). In addressing such issues, we will favour an approach that preserves the existing columns “as is”, but could add new columns for extra annotations and/or metadata (e.g., alternate class assignment). The same treatment could be used for other eventual issues identified by the community (i.e., beyond class ambiguity).
>
> We augmented our maintenance plan in the paper to highlight this approach for the next updates. Topic classification results with BERT are now also included in the appendix for reference.
>
> **We provide a further response in the comment below.**

---

### Official Review · Reviewer_7v8M · 2024-07-25
**a novel dataset designed to evaluate large language models on unseen content**

**Rating:** 5
**Confidence:** 3
**Correctness:** The claims seem correct to me.
**Clarity:** It is well written.

**Review:**

Pros:
1. This paper addresses a significant challenge in the evaluation of LLMs by providing a dataset that mitigates the issue of data contamination.

2. The authors detail a thorough process for creating reference documents and associated question-answer pairs. This includes human content creators and annotators, which enhances the dataset's quality and relevance.

3. The paper presents extensive benchmarking experiments using a wide range of state-of-the-art LLMs.

Cons:

1. While the use of third-party vendors for data annotation is practical, it introduces variability and potential quality issues. Although quality control measures are in place, the dependence on external providers may affect the dataset's consistency.

2. Potential for Data Leakage: Despite efforts to prevent data leakage, the gradual release of dataset splits may still pose a risk, especially if the released data is used in training future models.

3. Evaluation Limitations: The paper acknowledges the difficulty and expense of running multiple trials for benchmarking. This limitation affects the robustness of the experimental results, as error bars and variance measures are not reported.

**Strengths:**

1. Clear Contribution and Methodology: The paper clearly outlines its contributions and the methodology used for dataset creation, which adds to the transparency and reproducibility of the research.
2. Scalability and Accessibility: By releasing the dataset in splits over time, the authors ensure continued relevance and prevent immediate data leakage. This strategy enhances the longevity and utility of the dataset for future research.

**Additional Feedback:**

See above.

**Documentation:**

They are providing a huggingface url which has dataset they curated.

**Ethics:**

no ethical concerns

**Limitations:**

yes, the authors have addressed the limitations.

**Opportunities For Improvement:**

Please see the above review.

**Relation To Prior Work:**

Yes

**Summary And Contributions:**

The paper introduces a novel dataset designed to address the challenge of data contamination in the evaluation of LLMs. The dataset, named REPLIQA, consists of five splits, four of which have not been exposed to the internet prior to this publication. Each sample in REPLIQA includes a reference document crafted by human annotators, a question about the document's topic, a ground-truth answer derived directly from the document, and the paragraph containing the answer. The dataset is intended for use in question-answering and topic retrieval tasks, ensuring that models are evaluated on truly unseen content.

---

> ### Author Rebuttal · Authors · 2024-08-16
>
> We thank the reviewer for their comments which we believe help improve the manuscript. We are glad the reviewer found that we address a significant challenge in the evaluation of LLMs, appreciates the quality and relevance brought by our thorough dataset-creation process, and acknowledges our extensive benchmarking experiments using a wide range of state-of-the-art LLMs.
>
>   - **use of third-party vendors for data annotation…introduces variability**
>
> We completely agree with the reviewer that such a large scale data annotation effort poses risks and requires careful quality control. We highlight that several control measures were taken in order to minimize quality issues, which resulted in a fairly clean dataset. Namely, quality controls were carried out by the annotators in a peer review manner, and further detailed quality inspections were performed in a sample of the data by writers not involved in the annotation process. Moreover, after the end of the annotation procedure, further checks were performed by us to detect unnatural text.
>
> We refer the reviewer to Appendix D for details concerning the annotation process and the instructions given to annotators, including the various quality checks performed by the vendor and annotators. Appendix B highlights issues we personally identified and rectified. Appendix C.1 transparently discloses further potential irregularities we observed but left unaltered: less than 1% of the dataset is concerned, and we believe that releasing these samples verbatim will help the community better understand the dataset in the future.
>
> We finally mention that the Hugging Face dataset page linked in the paper (https://huggingface.co/datasets/ServiceNow/repliqa) has a very useful data preview tool that enables easy visualization of the data directly on the browser, so that quality issues can be easily assessed. The page also supports issue reporting, and any reported quality concern will be treated by the repo’s maintainers: confirmed quality problems will be dutifully documented, and major issues may be removed.
>
>   - **the gradual release of dataset splits may still pose a risk, especially if the released data is used in training future models**
>
> Thank you for mentioning that. Our manuscript already acknowledges that our solution isn't perfect, stating that "we have opted for merely delaying the risk of leakage by staggered dataset releases". We are not aware of any perfect alternative that allows an open distribution of the dataset while forbidding leakage in models' training sets worldwide. The reviewer may have in mind the alternative option of a single test set that would remain "secret" forever: in that scenario, people would upload their models for us to benchmark, and we would report the results summary. But this kind of approach has its own issues, to name a few: there is no transparency, there are substantial costs involved, and not everyone is ready to upload their model to an inference endpoint that they don't physically control. Instead, our preferred "staggered releases" solution favours transparency, simplicity and flexibility.
>
> That being said, we are confident in the "leak resistance" of our benchmark for any model released before June 9th 2025 (the date of our last split's release). Our five splits are identically distributed by construction, and the same synthetic document cannot appear in different splits. Training on RepLiQA_0 today may confer a minor advantage on RepLiQA_1 performance, but that advantage would be solely due to the model learning the “style” of RepLiQA’s documents, questions and answers: no “fake facts” would leak.
>
> Moreover, models that were never trained on any RepLiQA splits should perform similarly well on all RepLiQA splits. This means that if a model performs significantly better on an “old” RepLiQA split than on new ones, that model was likely trained on the old split. Our staggered release schedule may thus be seen as an experiment enabling the study of interesting questions. Which models trained on RepLiQA? At what rate does such leakage occur?
>
> We also remark that sequential releases offer an opportunity to improve the dataset over time, incorporating feedback from the community in between releases.
>
>    - **The paper acknowledges the difficulty and expense of running multiple trials for benchmarking. This limitation affects the robustness of the experimental results, as error bars and variance measures are not reported.**
>
> Indeed, evaluating such a large number of state-of-the-art language models using a large evaluation benchmark incurs a relevant cost, and running multiple trials becomes infeasible. Due to the standard errors for recall being too small to be readable in the figures (0.25% to 0.28% for RepLiQA and 0.29% to 0.50% for TriviaQA), we added them to the associated text to address the reviewer’s concern.  We also prepared a new plot (it can be seen in the pdf file attached to this rebuttal) showing a clear shift in performance metrics distributions across RepLiQA and TriviaQA. We note that our main claim is that Q&A evaluations in datasets that cover data available on the web  are confounded by model memory. We believe this plot makes the evidence of this being the case more clearly observed. Thank you for the suggestion.

---

> > ### Author Response · Authors · 2024-08-23
> >
> > Dear reviewer, thank you again for your time and feedback. While we strived to address all of the reviewer's concerns and believe to have done so, kindly let us know if there's any further question you might have or if there are any issues left unaddressed. We will be happy to continue the discussion and provide further details.

---

### Author Rebuttal · Authors · 2024-08-16

We thank all reviewers for their insightful comments. We believe those improved our manuscript significantly. We would like to briefly summarize new results, experiments, and additions to the manuscript done during the rebuttal phase:

  - We added results of a topic classifier trained on RepLiQA. We achieve a very high accuracy on that task (+95% accuracy), which suggests there is not relevant topic ambiguity within the dataset. Code to reproduce our experiments can be inspected and executed directly on browser using the following notebook: https://colab.research.google.com/drive/1l6TgMzlo3X9Ev4bfhsvN8-eA8MkBeMk-?usp=sharing.

  - We added results obtained from a model fine-tuned on QA tasks. Namely, we evaluated nvidia/Llama3-ChatQA-1.5-8B on both RepLiQA and TriviaQA. Code implementing inference for this model can be found at https://github.com/ServiceNow/repliqa/blob/main/hf_qa_eval.ipynb.

  - We expanded the Limitations section with a discussion focused on the risks due to the annotations being carried out by crowdsourced workers.

  - We added extra plots to make the gaps across datasets in figure 4 clearer, to highlight the discrepancy between evaluations on the two datasets we considered, and to show low topic ambiguity.

  - We clarified that the dataset is distributed under the CC BY 4.0 license (as is currently indicated on Hugging Face). The previous version of the manuscript incorrectly stated CC BY-NC 4.0 .

All changes will be reflected in the camera-ready version. Besides those changes and additions, detailed responses were provided to each reviewer. Once again, thank you for the useful and actionable feedback.

---

### Decision · Program_Chairs · 2024-09-26

**Decision:**

Accept (Poster)

**Comment:**

This is a new dataset of 90k question-answer pairs on 18k reference documents that have never appeared on the internet before. The dataset was produced by hiring human content writers to invent reference documents in imaginary scenarios and create question-answer pairs that can only be answered using these novel documents.
This is an important and valuable  research effort since this new dataset has not appeared on the internet before and can be used to test contamination issues, that are currently a major problem in LM evals.
The authors did also a good job in responding to reviewer concerns in their rebuttal and I think this paper deserves to be published.